# The ctenophore Mnemiopsis leidyi deploys a rapid injury response dating back to the last common animal ancestor
Dorothy G. Mitchell [1,2], Allison Edgar [1], Júlia Ramon Mateu[1], Joseph F. Ryan [1,2] & Mark Q. Martindale [1,2] ✉

Regenerative potential is widespread but unevenly distributed across animals. However, our understanding of the molecular mechanisms underlying regenerative processes is limited to a handful of model organisms, restricting robust comparative analyses. Here, we conduct a time course of RNA-seq during whole body regeneration in *Mnemiopsis leidyi* (Ctenophora) to uncover gene expression changes that correspond with key events during the regenerative timeline of this species. We identified several genes highly enriched in this dataset beginning as early as 10 minutes after surgical bisection including transcription factors in the early timepoints, peptidases in the middle timepoints, and cytoskeletal genes in the later timepoints. We validated the expression of early response transcription factors by whole mount in situ hybridization, showing that these genes exhibited high expression in tissues surrounding the wound site. These genes exhibit a pattern of transient upregulation as seen in a variety of other organisms, suggesting that they may be initiators of an ancient gene regulatory network linking wound healing to the initiation of a regenerative response.

Regeneration in animals is the capacity to regrow and repattern lost tissues and structures. Most animal phyla contain lineages that possess some potential to regenerate, ranging from the ability to replace specified tissues to entire body regions, while also containing lineages that lack regenerative potential[1,2]. This uneven distribution of regenerative capacity has made it difficult to identify a clear pattern of gain and loss across animals or whether there is an ancestral regeneration program common to all animals. However, it should be noted that most members of the earliest branching animal clades have a high capacity for whole-body regeneration (WBR). Identifying specific mechanisms of regeneration in disparate animal lineages will provide insight into how many times regenerative abilities arose and how they subsequently evolved. If there is an ancestral regeneration program, there are likely core components that have been conserved across animal lineages. Conversely, if mechanisms of regeneration are lineage-specific, identifying the components that are required for successful regrowth will provide important insight into the molecular basis for the stability of cell fate across animal evolution. The regulation of gene expression during regeneration is of particular interest because many of the genes involved are also deployed during embryogenesis and post-embryonic growth, and an understanding of the context-dependent differences in their regulatory interactions is just emerging[3].

Regeneration in most animals can be divided into three phases: (1) physical closure of the wound following an injury, (2) accumulation of cells competent to replace missing cell types either through the proliferation of precursor cells or transdifferentiation, (3) differentiation and morphogenesis of cells to accurately replace missing tissues and structures[4–7]. Using this framework, we can determine if the molecular profiles associated with each of these steps are shared between animal lineages. Similarities in gene expression could ultimately be the result of ancestral homology or convergent homoplasy, but in either case, they pose an exciting opportunity to uncover the properties that drive the emergence of regenerative capabilities[5,8].

Just as the comparison of gene regulatory networks involved in early embryogenesis revealed similarities across distantly related species[9–13], gene regulatory networks linking the early wound response with later regenerative events are being built in multiple model systems using comparative gene expression, chromatin accessibility data, and functional manipulations[4,14–17]. These studies have demonstrated that orthologs of wound response genes (e.g., bZIP transcription factor genes) modulate downstream gene expression (Wnt signaling pathway members) required for regeneration widely across the animal tree[5]. These regulatory connections suggest that early injury responses associated with wound healing are

[1] Whitney Laboratory for Marine Bioscience, University of Florida, Saint Augustine, FL, USA. [2]Department of Biology, University of Florida, Gainesville, FL, USA. ✉e-mail: mqmartin@whitney.ufl.edu

not disconnected from regeneration but rather that they are integral in its initiation. However, it remains unknown if these similarities in wound response gene expression are part of an ancestral gene regulatory network of regeneration.

To gain insight into possible ancestral gene regulatory states for regeneration, we have focused on Ctenophora, the sister group to the rest of animals[18–22]. Also known as comb jellies, ctenophores are a clade of gelatinous marine invertebrates that have been the subject of developmental and regenerative studies for over a century[23–30]. The ctenophore *Mnemiopsis leidyi* exhibits extensive whole-body regeneration, which can be effectively studied in a laboratory environment (e.g.,[24,31–33].). The body plans of most ctenophores, including *M. leidyi*, exhibit rotational symmetry with numerous distinct structures that can be surgically removed and scored for regeneration[34]. Notable structures, all of which can be regenerated, include two tentacle bulbs, one gravity-sensing aboral organ, 8 longitudinal comb rows containing giant ciliary ctene plates, ciliary grooves connecting comb rows to the aboral sense organ, endodermal canals running underneath the comb rows and tentacle bulbs and around the pharynx, and a pharynx that connects the endodermal gut to the mouth opening on the oral pole (Fig. 1). Although morphologically distinct from the older "lobate" stage, the younger "cydippid" stage of *M. leidyi* is considered a functional adult containing all permanent structures[35], and the ability to regenerate is

maintained throughout its life after a post-embryonic onset[24,29]. We chose to conduct our study on cydippid stage animals of ~1–3 mm diameter, which is favorable for surgical manipulations, imaging, and high-quality RNA extraction. *M. leidyi*'s whole body regeneration can be broken into three phases: wound healing occurs within 2 h; cell proliferation begins at around 6 h, and regeneration of all missing structures and cell types is completed by 48 h after injury. Furthermore, this regeneration is accomplished without an apparent blastema or any detectable scarring[36]. Here, we use bulk RNA-seq to identify the transcriptomic hallmarks of each phase of regeneration, following the same time course.

Using a pairwise differential gene expression approach, we generated a primary dataset specifying distinct sets of upregulated and downregulated genes between sequential time intervals. To gain insight into the molecular processes associated with the regulatory changes detected across the time course, we tested each of these lists for gene ontology enrichment. In an effort to filter the dataset to search for candidate genes, we combined the results from three differential gene expression methods and generated a high-confidence list of differentially expressed genes (DEG) regulated dynamically between any of the subsequent time points in the time course. We identified significant changes in gene expression by both RNA-seq, corroborated by RNA in situ hybridization, as early as 10 min after injury. We found that *M. leidyi* deploys several genes identified as part of the early

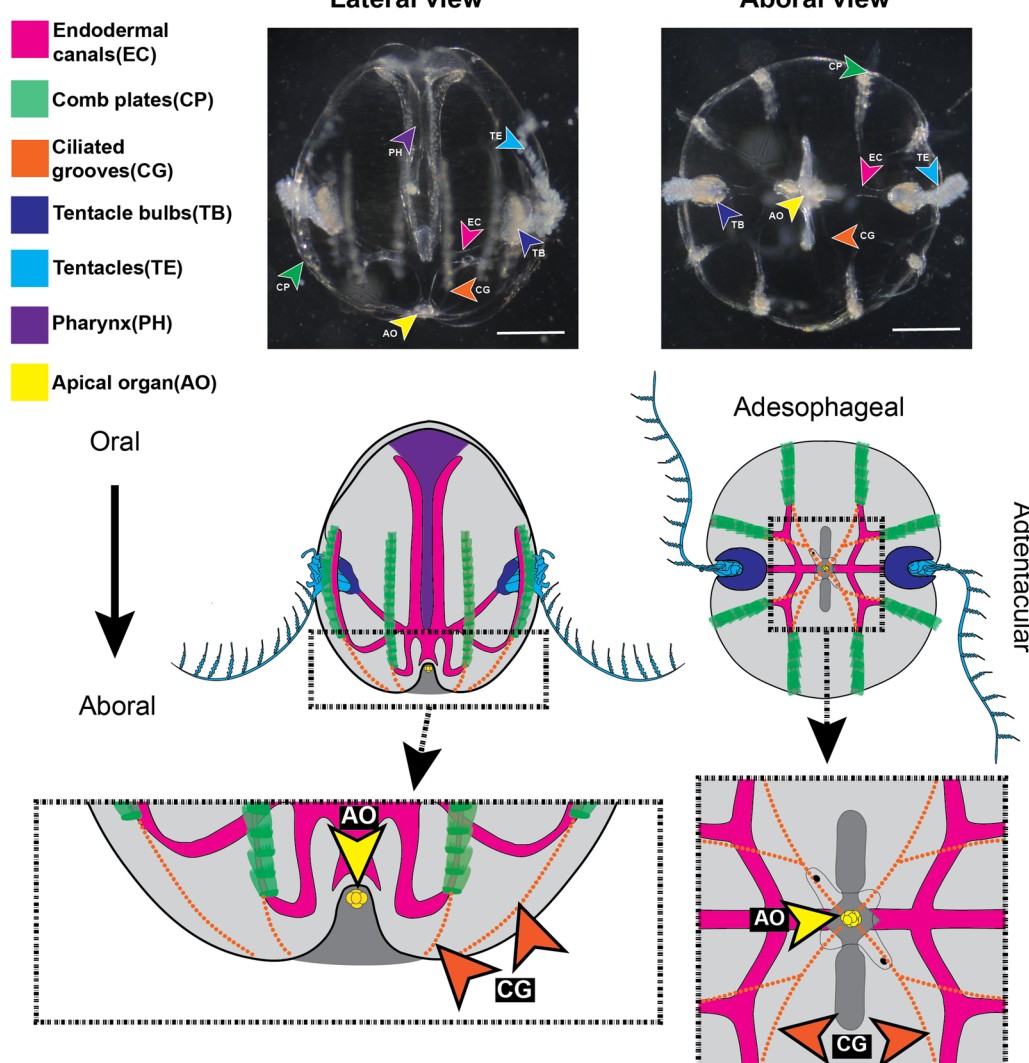

**Fig. 1 | Anatomy of *Mnemiopsis leidyi*.** Structures are color-coded as follows: Pink = Endodermal canals (EC), Green = Comb plates (CP), Orange = Ciliated grooves (CG), Dark Blue = Tentacle bulbs (TB), Light Blue = Tentacles (TE), Purple = Pharynx (PH). Scale = 500 μm.

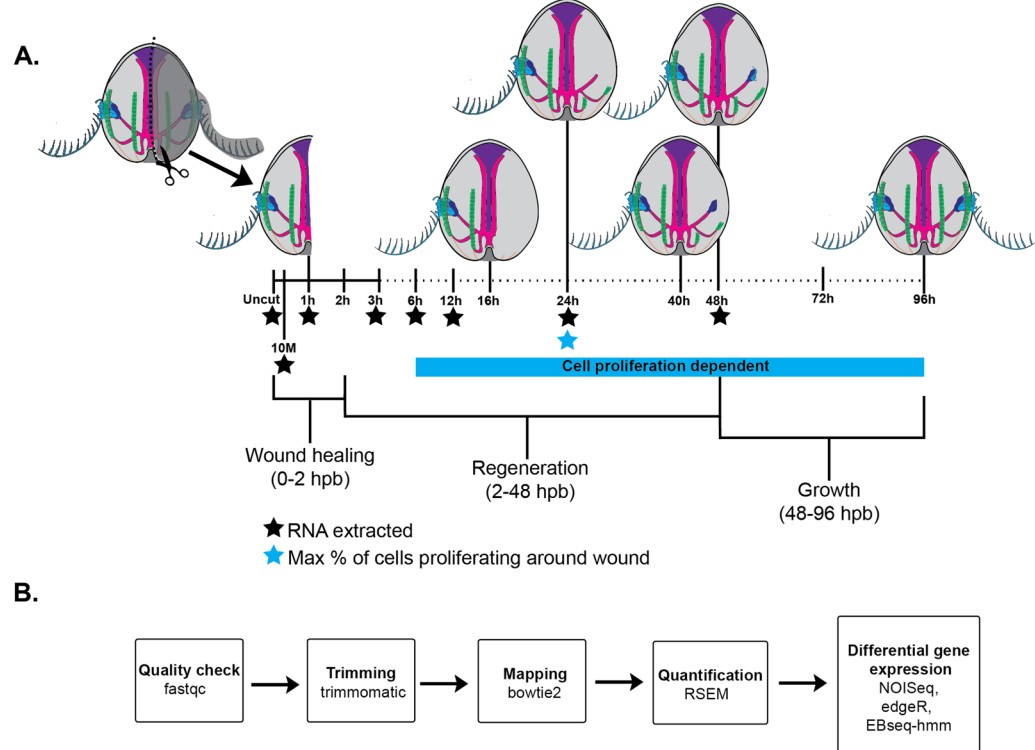

**Fig. 2 | Summary of methods for transcriptomic analysis. A** Experimental design for surgeries and RNA extraction across regenerative timeline. **B** Bioinformatic pipeline following bulk RNA-seq. Uncut, 10 m = 10 min post bisection (mpb), 01 h–48 h = hours post bisection (hpb).

injury response in other animal lineages, including transcription factors containing bZIP and ETS motifs. Taken together with what is known of the molecular events of regeneration in other well-studied model organisms, these results show that we have identified a set of injury response genes upstream of regeneration in the earliest animals, which could represent an ancient gene regulatory network promoting total body regeneration.

## Results
### Gene ontology of DEG associates each phase of regeneration with molecular processes

We performed longitudinal bisections[36] and extracted total RNA from three biological replicates, each containing 20–30 individuals per sample. Sampled time points included uncut control, 10 minutes post bisection (mpb), and 1, 3, 6, 12, 24, and 48 hours post bisection (hpb) (Fig. 2A). First, we used the differential gene expression program NOISeq[37] (v2.44.0) to identify genes which were upregulated or downregulated between sequential time points, resulting in gene expression profile transitions for each sequential time interval (e.g., 3hpb–6hpb) (Fig. 2B, Supplementary Fig. 2A). We chose to use this non-parametric and pairwise program as our primary method of DEG analysis due to its favorability for the number of biological replicates (3) and time intervals (7) in our study[37,38]. We grouped sequential time intervals and designated them into three temporal phases: early (uncut-10mpb,10mpb–1hpb), middle (1–3hpb, 3–6 hpb), and late (6–12hpb,12–24hpb, 24–48hpb). The early phase, which corresponds to the acute injury response before the completion of wound healing, is the phase with the fewest number of DEG per timepoint (Supplementary Fig. 2A). The middle phase, which covers the closing of the wound up to the onset of cell proliferation, exhibits the highest number (Supplementary Fig. 2A). The late phase includes the peak of cell proliferation around the wound site through patterning and regrowth of removed structures and includes fewer DEG (Supplementary Fig. 2A). However, we recognize that the number of DEG in each phase undoubtedly is influenced by the intervals between chronological time points, with the earlier phases being more densely sampled.

We separated the lists of genes from each time interval into upregulated or downregulated DEG to identify the enrichment of gene ontology (GO) in each direction of regulation (i.e., up or down). Using TOPGO (v2.46.0) for enrichment followed by redundancy reduction of significant terms using REVIGO (v1.12.0), we generated GO term associations to each interval (Fig. 3). As the *M. leidyi* gene and protein models share identifiers (e.g., ML12345a), we performed a reciprocal best BLAST (blastp, e-value < 0.001) of all of the *M. leidyi* protein models against the NCBI nonredundant human protein database (2020). As a result, 9,991 out of the total 16,548 *M. leidyi* protein models received a BLAST hit, thus allowing us to annotate *Mnemiopsis* gene IDs associated with the GO enrichment as well as the other methods implemented throughout our study (Supplementary Data 3). This analysis revealed top GO terms associated with upregulation and down-regulation in the sequential intervals of the time course (Supplementary Data 6).

In summary, the early timepoints feature enrichment for chitin binding (GO:0008061), DNA-binding transcription factor activity (GO:0003700), and metalloendopeptidase activity (GO:0004222) in the upregulated DEG and calcium ion binding (GO:0005509) in the downregulated DEG (Fig. 3A, Fig. 4B, Supplementary Fig. 1A, B). The middle timepoints include structural molecule activity (GO:0005198) and G-protein coupled receptor activity (GO: 0004930) in upregulated DEG and RNA binding and catalytic activity in downregulated DEG (Fig. 3B, Fig. 4C, Supplementary Fig. 1C, D). The late timepoints feature structural molecule activity (GO:0005198) in upregulated DEG while the structural constituent of the cytoskeleton (GO:0005200) was identified in the downregulated DEG from 12hpb–24hpb while also being enriched in the upregulated DEG in 24hpb–48hpb (Fig. 3C, Fig. 4D, Supplementary Fig. 1E).

Next, we extracted the DEG contained in the enriched GO terms from each phase, allowing us to isolate groups of genes and interrogate their dynamic regulatory patterns assigned from NOISeq (e.g., uncut–10mpb up, 1–3hpb down) across the entirety of the time course (Fig. 3D, Supplementary Data 6). In particular, we examined the transcription factor genes under DNA binding transcription factor binding activity (GO:0003700)

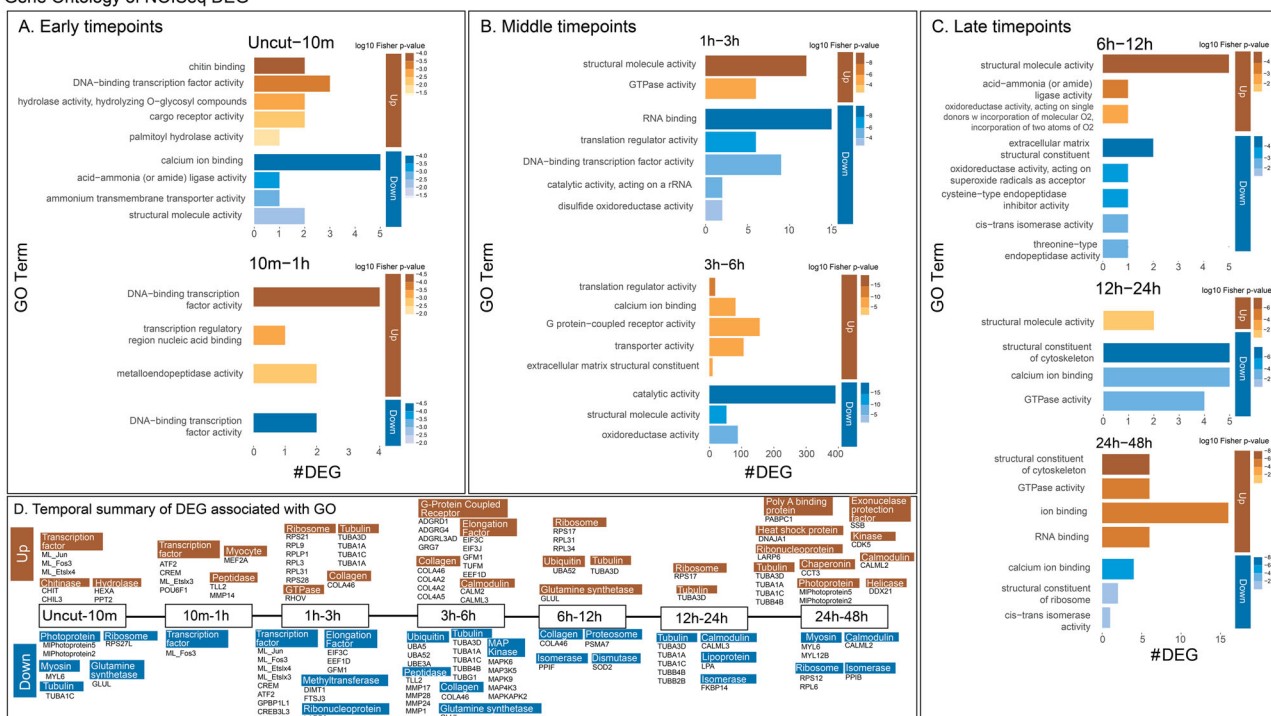

**Fig. 3 | Gene Ontology enrichment in intervals across the regenerative timeline.**
**A** Early timepoints (Uncut-10, 10 min–1 h) **B** Middle timepoints (1 h–3 h, 3 h–6 h)
**C** Late timepoints (6 h–12 h, 12 h–24 h, 24 h–48 h). The total # of DEG in each term
in each grouping is along the *x*-axis, the GO terms are stacked on the y-axis, and
*p*-value is indicated by the color intensity in (**A**–**C**). **D** Summary of gene ontology
enrichment analysis across the entire time course.

enriched in the early and middle phases (Fig. 4B). Peptidase genes were
identified under metalloendopeptidase activity (GO:0004222) and catalytic
activity (GO:0003824) enriched in the early and middle phases (Fig. 4C).
Finally, alpha-tubulin genes in structural constituent of cytoskeleton
(GO:0005200) enriched in the late phase of the time course (Fig. 4D).

**Transcription factors are upregulated in the early and middle phases of the regenerative timeline**

Transcription factors directly control gene expression levels by binding to
regulatory DNA sequences. The cascades of such transcription factors are
fundamental to gene regulatory networks which control diverse processes,
such as development and regeneration[16,39]. The GO term DNA binding
transcription factor binding activity (GO:0003700) was enriched in the DEG
upregulated uncut-10mpb and includes the Fos proto-oncogene
(ML_Fos3 – ML182032a), the Jun proto-oncogene (ML_Jun –
ML1541120a), and an E26 transformation specific (ETS) domain-
containing gene (ML_Etslx4 –ML282527a) (Fig. 4B). This term was also
enriched in the DEG upregulated 10mpb-1hpb and included the CAMP
responsive element modulator (CREM – ML077623a), the activating
transcription factor 2 (ATF2 – ML057318a) and a different ETS domain-
containing gene (ML_Etslx3 –ML09109a) (Fig. 4B). Upregulation in the
early phase indicates that the expression of these genes is not controlled by
de novo protein synthesis of an upstream regulator but rather a signaling
cascade that activates existing regulators, such as a temporary MAP kinase
cascade[40]. Genes that exhibit such rapid changes in gene expression in
response to a stimulus are often called "early response genes"[41,42] or
"immediate early response genes"[5,43].

Rapid, transient expression of such early response genes has been
identified following injury in diverse animal contexts[15,44–50]. The protein
products of early response genes are thought to activate downstream targets
in the gene regulatory network, which are thus called "secondary response
genes"[51] and exhibit slightly later gene expression changes. The DNA
binding transcription factor activity term was also enriched in the DEG
downregulated 1hpb–3hpb and included all of the aforementioned

transcription factor genes in addition to the CAMP Responsive Element
Binding Protein 3 Like 3 (CREB3L3–ML015722a) and the Activating
Transcription Factor 6 Beta (ATF6B–ML08021a). In addition to being
downregulated 1–3hpb, CREB3L3 (ML015722a) and ATF6B (ML08021a)
are both also upregulated 3-6hpb, suggesting that their regulation is delayed
(Fig. 4B). However, their upregulation is considerably later than the early
response genes, it is likely that they are regulated by something in between
(e.g., genes upregulated 1–3hpb)

**Peptidase transcripts are downregulated prior to the onset of cell proliferation**

Matrix metalloproteinases (MMPs) function in the degradation of extra-
cellular matrix (ECM) components (e.g., collagen). Here, we follow the
recommended nomenclature and refer to all proteolytic enzymes (i.e.,
protease, peptidase, and proteinase) as peptidases for their involvement in
the breakdown of peptide bonds at various locations across polypeptide
chains[52]. Included in the enrichment of Metalloendopeptidase activity
(GO:0004222) in the DEG upregulated 10m-1hpb are metalloproteinase 14
(MMP14–ML305524a) and tolloid-like 2 (TLL2–ML007435a) genes
(Fig. 3B, Fig. 4C, Supplementary Data 6). These genes are also found in the
catalytic activity (GO:0003824) term enriched in the DEG downregulated at
3hpb–6hpb along with metalloproteinase 24 (MMP24–ML14875a),
metalloproteinase 1 (MMP1–ML14713a), metalloproteinase 28 (MMP28 –
ML33825a), metalloproteinase 28 (MMP28–ML13379a), metalloprotei-
nase 28 (MMP28–ML073224a), and caspase 3 (CASP3–ML154125a)
(Fig. 3B, Fig. 4C, Supplementary Data 6). While only MMP14 and TLL2 are
upregulated in the early phase, during the time of wound closure, all are
included in the enrichment of catalytic activity in the downregulated DEG
during 3hpb–6hpb prior to the onset of cell proliferation (Fig. 3B, Fig. 4C,
Supplementary Data 6).

**Tubulin genes are upregulated late in the regenerative timeline**

Tubulin proteins are essential for diverse functions, including both cell
growth and migration, depending upon their interactions with one another

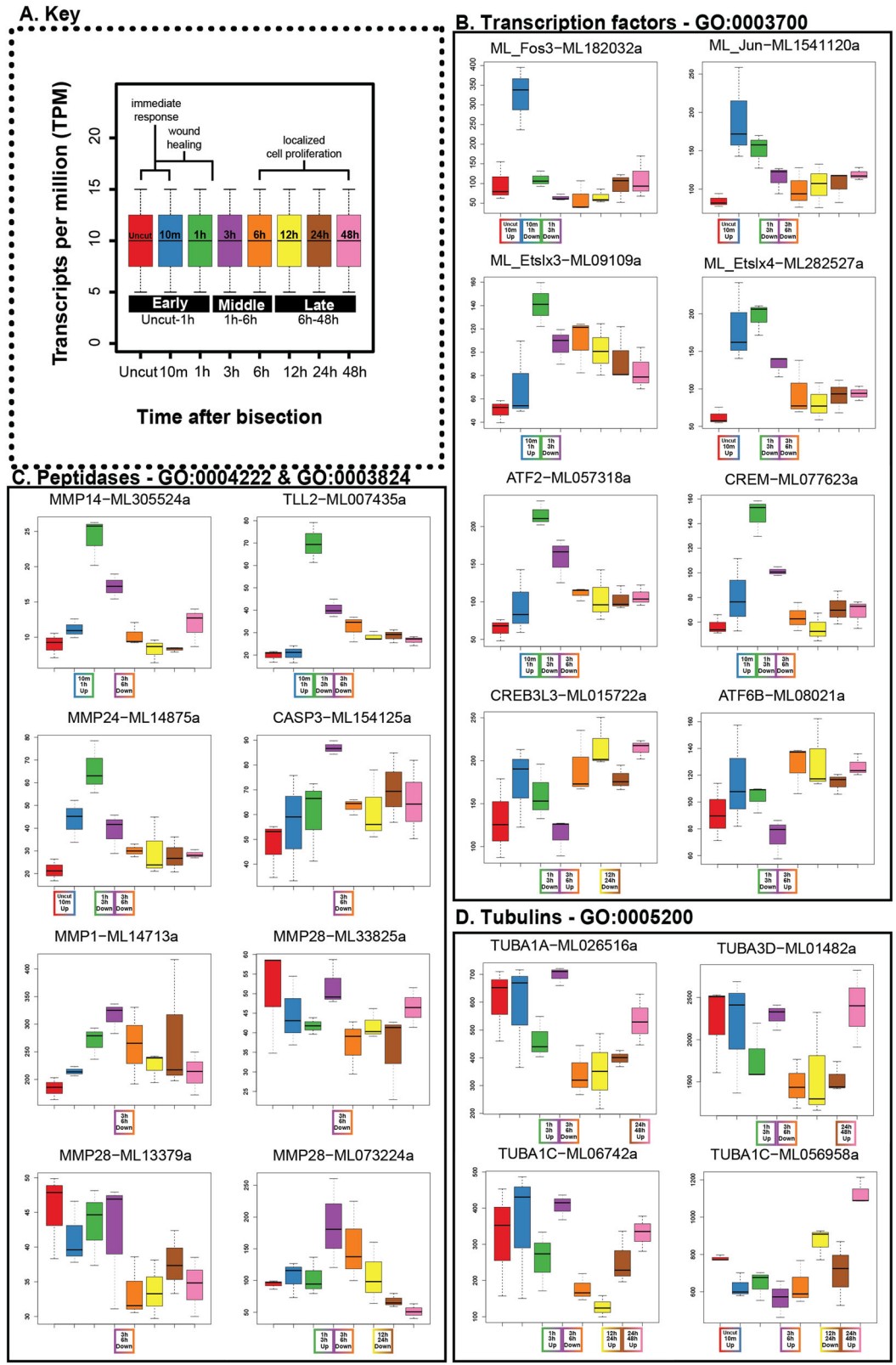

**Fig. 4 | Expression of differentially expressed genes from significantly enriched GO categories.** Gene nomenclature is based on the best BLAST hit to the human protein database unless we performed phylogeny (see Supplementary Figs. 5 and 6). **A** Key panel that demonstrates color coding, labeled axis, and relevant biological events covered over the time course. The time points (*x*-axis) are shown in this panel as uncut, 10 min, 1 h, 3 h, 6 h, 12 h, 24 h, 48 h (min = minutes post bisection, h = hours post bisection); expression level (*y*-axis) is in transcripts per million (TPM). **B** Transcription factors in GO:0003700. **C** Peptidases in GO:0004222 and GO:0003824. **D** Alpha tubulin genes in GO:0005200.

and other proteins, as well as post-translational modifications[53]. The GO term structural constituent of the cytoskeleton (GO:0005200) contains several alpha-tubulin genes. This term is enriched in the DEG upregulated 24–48hpb and includes tubulin alpha 1a (TUBA1A–ML026516a), tubulin alpha 3d (TUBA3D–ML01482a), tubulin alpha 1c (TUBA1C–ML06742a), and tubulin alpha 1c (TUBA1C–ML056958a) (Fig. 3C, Fig. 4D). Interestingly, TUBA1A (ML026516a), TUBA3D (ML01482a), and TUBA1C (ML06742a) are also upregulated 1–3hpb and subsequently downregulated 3–6hpb (Fig. 4D, Supplementary Data 4). The biphasic, coordinated expression of these three alpha-tubulin genes suggests their activation both in the middle and late phases of the regeneration program.

### Overlapping of DEG methods condenses DEG into candidates for experimental validation

The overlapping of results from at least three differential gene expression programs is highly effective in the reduction of false positives[38]. As such, we generated a conservative list by merging the DEG found across all subsequent time points (i.e., uncut–10 min, 10 min–1 h, etc.) from three methods: nonparametric pairwise NOISeq[37] (v2.44.0), parametric pairwise edgeR[54] (v3.42.4), and full time-course Bayesian estimation for ordered experiments EBseq-hmm[55] (v1.34.0) (Fig. 2B, Supplementary Data 5). This allowed us to condense our results into the highest-confidence DEG (Supplementary Fig. 2D). Each method identified distinct but overlapping sets of genes as differentially expressed (Supplementary Fig. 2, Supplementary Data 5). Temporal patterns of differential expression found by the two pairwise (NOISeq and edgeR) algorithms are similar, with the greatest number of DEG identified at 3-6 hpb (Supplemental Fig. 2A-B), which is generally consistent with the ordered analysis of EBseq-hmm in which the greatest number of genes were categorized into the expression path "Up–Down–Down–Up–Down–Down–Up", corresponding to the ordered time intervals (e.g., uncut–10mpb,10mpb–1hpb), indicating that the majority of DEG were regulated 3–6 hpb (Supplementary Fig. 2). The overlapping set includes 118 genes which we refer to as the consensus DEG (Fig. 4A). We identified these genes using the *M. leidyi* protein models; 72/118 received a best BLAST result to the human genome (Supplementary Fig. 3A).

### Early response genes feature bZIP and ETS domain family transcription factors

Using cluster membership and regulatory designations from NOISeq, we identified putative early-response genes in the consensus set of DEG. Cluster 1 and cluster 2 contained the bZIP-domain containing ML_Fos3 (ML182032a) and ML_Jun (ML1541120a), while cluster 2 also contains ML_Etslx4 (ML282527a), which contains the ETS binding domain. These three early response gene candidates are also included in the enriched GO term DNA-binding transcription factor activity in the DEG upregulated in uncut-10mpb (Fig. 3A, Fig. 4B, Supplementary Data 6). We then validated the identity of these early response genes by generating a molecular phylogeny of their DNA-binding domains.

Genes containing the basic-region leucine zipper (bZIP) domain form a large transcription factor family and affect diverse cellular processes across eukaryotes[56–58]. This domain is highly conserved across animals and consists of leucine-rich repeats that aid in the dimerization of these proteins into a functional transcription factor[59,60]. We searched the protein sequence database of *M. leidyi*, *Drosophila melanogaster*, human and *Nematostella vectensis* using a hidden Markov model (HMM) to identify genes containing the bZIP domain across these species (pfam: PF00170, bzip_1). Based on our search, six of the seven *M. leidyi* transcription factor genes identified in the DNA-binding transcription factor activity GO category enriched early in the time course contain the bZIP domain: ML_Jun (ML1541120a), ML_Fos3 (ML182032a), ATF2 (ML057318a), ATF6B (ML08021a), CREM (ML077623a), and CREB3L3 (ML015722a) (Fig. 4B, Supplementary Data 7). We constructed a maximum-likelihood tree to infer relationships of bZIP-containing genes across these 4 species. The tree shows that the *M. leidyi* genome contains a single Jun ortholog that we have called ML_Jun

(ML1541120a) and three Fos genes that we have named ML_Fos1 (ML09433a), ML_Fos2 (ML09961a), and ML_Fos3 (ML182032a) (Supplementary Fig. 5, Supplementary Data 7).

The remaining gene in the transcription factor GO category enriched early is ML_Etslx4 (ML282527a), which contains the E26 transformation specific (ETS) domain (Fig. 3A, Fig. 4B). This highly conserved domain follows a winged-helix-turn-helix configuration and genes containing it exhibit diverse gene regulatory functions[61]. We used the same search strategy as above to search for this domain across the same 4 species (pfam: PF00178, ets). ML_Etslx4 (ML282527a) and ML_Etslx3 (ML09109a) were found in this search as well as a second homolog each, ML_Etslx6 (ML46087a; BS = 77) and ML_Etslx1 (Ml10621a; BS = 86) within the *M. leidyi* genome (Supplementary Fig. 6, Supplementary Data 7).

### Early-responding transcription factors are expressed proximate to the site of injury

To verify the rapid temporal expression changes seen in the candidate early response genes, we used whole-mount, colorimetric RNA in situ hybridization (ISH) to examine their spatial expression in regenerating cydippids. We repeated the bisections as for the RNAseq analysis and fixed animals after 10 min, 1 h, and 3 h[62]. Expression was detected for ML_Fos3 and ML_Jun RNA around the wound edge in the 10 mpb and 1 hpb time points but this signal was completely undetectable by 3 hpb (Fig. 5A, B, Supplementary Fig. 7). However, ML_Jun expression was also present in the tentacles and tentacle bulbs in all samples except the uncut control (Fig. 5B, Supplementary Fig. 7). We found that ML_Etslx4 expression is detectable at the 10 mpb and 1 hpb time point around the wound edge. The expression domain for ML_Etslx4 is clearly expanded beyond the wound site with signal apparent around the tentacle bulbs and under the comb rows (Fig. 5C, Supplementary Fig. 7). ML_Etslx4 mRNA became undetectable in the samples by 3 hpb (Fig. 5C, Supplementary Fig. 7). These ISH results validated the temporal expression patterns detected by RNAseq and showed that the spatial expression of these genes is largely restricted to the wound margin in the early injury response, although some expression is also seen in the tentacle bulbs.

## Discussion
### Transient expression of classic early response genes is highly conserved and likely connected to the onset of regeneration

The early response transcription factor genes identified in our analysis (i.e., ML_Fos3, ML_Jun, and ML_Etslx4) show upregulation within minutes following bisection, indicating that they are participating in an immediate injury response. The early upregulation of two bZIP transcription factors identified in this analysis (i.e., ML_Fos3 and ML_Jun) suggests they have a role in regulating gene expression in the subsequent phases of regeneration. *M. leidyi*'s rapid upregulation of these transcription factors following injury resembles other highly regenerative species. Fos mRNA is detected quickly and transiently around wounds in highly regenerative animals, including hydra[63], planarians[64], and earthworms[65], as well as in non-regenerative injury contexts such as following epithelial wounding in rat embryos[40,46]. In the anemone *N. vectensis*, Fos is upregulated 1 h following injury and downregulated by 4 h[44]. In *Hydra vulgaris*, chromatin accessibility around AP-1 (Fos/Jun dimer) transcription factor binding motifs increases from 0 to 3 h after amputation, indicating that genes upregulated during this time may be directly activated by this bZIP complex[15]. Although early response genes may operate on slightly different absolute timescales in different species[15,44,66], their rapid, transient expression following a stimulus remains strikingly consistent and likely coordinated for the initiation of the regenerative process. For these genes, sharp peaks of expression versus longer plateaus have functional consequences. Extended expression of AP-1 genes is associated with excess scarring in rat embryos, and the transient upregulation of these genes is critical for the proper migration and proliferation of keratinocytes[40,48]. Additionally, the application of the MAPK inhibitor U0126, which blocks regeneration in *N. vectensis*, also results in prolonged bZIP TF expression[44].

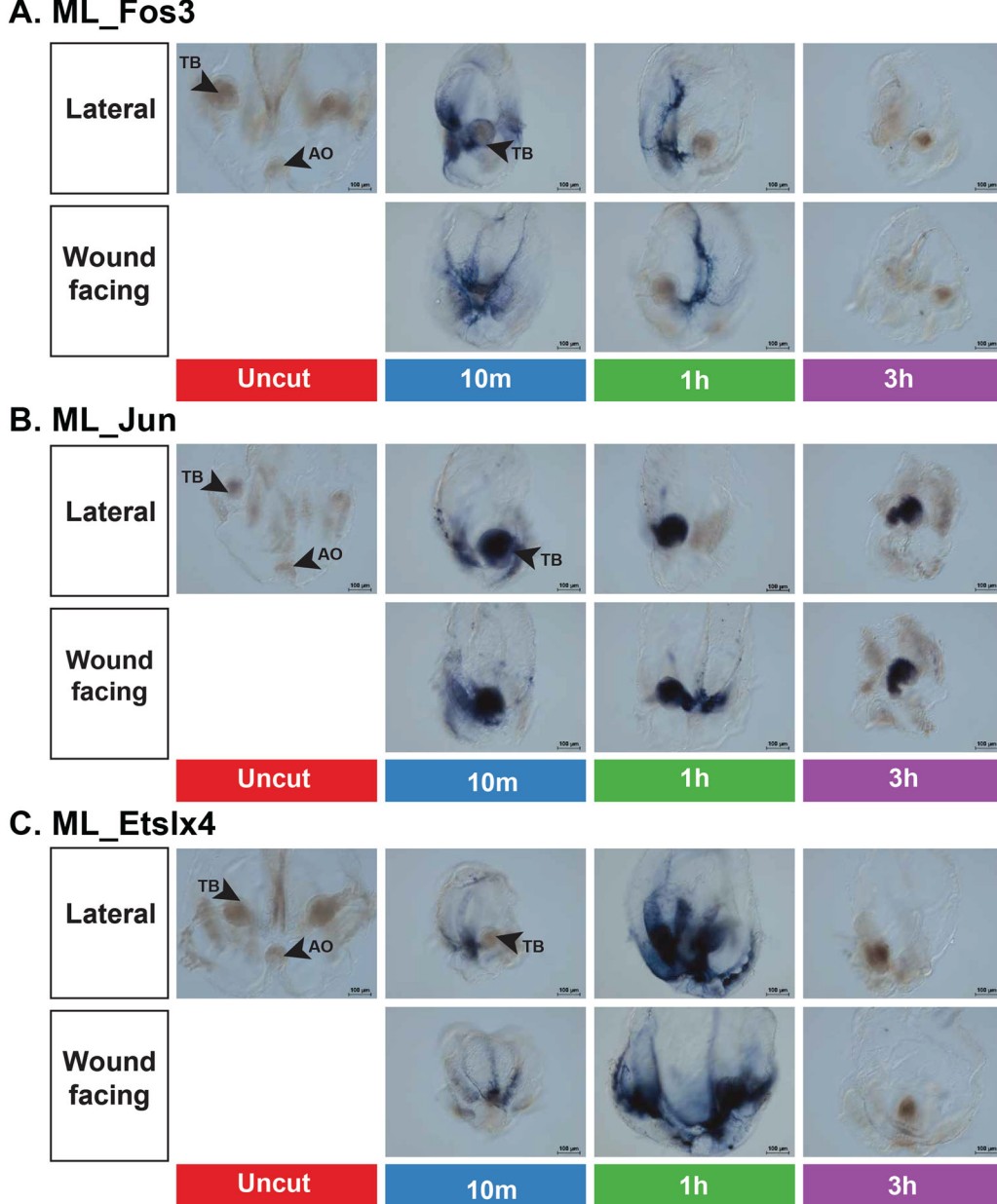

**Fig. 5 | Antisense RNA in situ hybridization following bisection.** **A** ML_Fos3 (ML182032a) (**B** ML_Jun (ML1541120a). **C** ML_Etslx4 (ML282527a) Lateral view shows tentacle bulb/s (TB) lateral to the aboral organ (AO) in uncut and cut animals. Animals are orientated with their oral side up. Wound-facing view orients the open site of bisection towards the viewer. Scale bar = 100 μm. Full good views of ISH assay and sense control probe results are in Supplemental Figs. 7 and 8, respectively.

Other components of our analysis also suggest regulatory control by transcription factors characterizes the early injury response in *M. leidyi*. The ETS domain-containing ML_Etslx4 could also be driving a regeneration-specific regulatory response. In *Drosophila* imaginal discs, the JNK/AP-1 pathway initiates Ets21C expression necessary for normal regeneration, and it is believed that together, these two transcription factors prolong the expression of other genes[50]. In our analysis, the early response genes (i.e., ML_Fos3, ML_Jun, and ML_Etslx4) were all upregulated uncut–10mpb, but ML_Etslx4 expression remains high at 1hpb, indicating that expression of this gene could be prolonged (Fig. 4B). It would be valuable to determine if these transcription factors form a coherent feedforward loop like that in *Drosophila*. These TFs could also initiate gene regulatory changes as pioneer factors. Pioneer transcription factors function in modifying gene regulation by recognizing binding site on closed chromatin, thus allowing access to transcriptionally silenced genes without substantial histone modification[67]. Therefore, TFs of this kind can prompt coordinated shifts in gene regulation, such as those observed in cell fate specification during development[68]. Coordinated control of regenerative mechanisms is also necessary for proper replacement of missing structures. Therefore pioneer transcription factor activity might be deeply conserved in the regenerative response. In *Hydra*, analysis of histone modification activity and enhancer structure following ectopic Wnt expression uncovered ETS-related TF binding motifs, suggesting their role as pioneer factors[17].

The mechanism by which these early response genes are initiated is unknown in *M. leidyi*. In other organisms, the influx of calcium and release of reactive oxygen species (ROS) (e.g., hydrogen peroxide) is one of the earliest responses to injury[69] deeply conserved across animals and plants[70]. ROS and/or calcium release localized to the wound site is essential for proper regeneration in *Drosophila*, zebrafish, *Hydra*, *Xenopus*, and planarians[71–75]. ROS release is also required for MAPK signaling in planarians[75] and JNK signaling in zebrafish[76]. It remains to be determined if ROS are necessary for regenerative function in *M. leidyi*. Our gene ontology analysis identified

rapid downregulation of calcium ion binding genes within 10mpb (Fig. 3A) These include, but are not limited to, two photoprotein genes previously characterized in *M. leidyi*[77] (Supplementary Data 6, Supplementary Fig. 2A).

## Peptidase transcripts are highly regulated in the scarless regeneration of *M. leidyi*

*M. leidyi* is capable of scar-free wound healing and completes regeneration without forming a blastema[36]. Scarless wound healing, discernable by the absence of excess collagen deposits that alter tissue texture, has been observed in several invertebrates[44,78] and a few vertebrates[79,80]. Regulation of collagen breakdown by peptidases is a key feature of scarless healing and regeneration. Peptidase activity has been observed or suggested in the regeneration of diverse animals. Matrix metalloproteinases (MMPs) exhibit gene upregulation following injury in *Hydra*, *Nematostella*, zebrafish, mouse, sponge, and earthworm[44,65,81–84]. The successful degradation of the ECM by MMPs is critical in the reduction of scarring and successful blastema formation during limb regeneration in several amphibian species[85,86], and scarring correlates with poor regenerative outcomes in these animals. It was found that in short-toe axolotl mutants, limb amputation results in excess scarring and undifferentiated blastemas, ultimately preventing limb regeneration[87]; MMP genes are not upregulated in short-toe mutants as well as non-regenerative Xenopus froglets[86]. Similarly, the application of an MMP inhibitor to the newt *Notophthalmus viridescens* prevents limb regeneration and results in excess scarring associated with an undifferentiated blastema[85]. In addition, the spiny mouse, *Acomys*, upregulates MMP while downregulating collagen following injury, whereas non-regenerative mice species do the opposite[88,89]. This points to the idea that there may be direct tradeoffs between scarring and regenerative success[90].

Peptidase genes were highly enriched in the GO analysis. In particular, a subset of the differentially expressed MMPs (MMP24–ML14875a, MMP14–ML305524a) and a tolloid-like gene (TLL2–ML007435a), all members of the metzincin superfamily of zinc-dependent proteases, are upregulated during the wound healing stage (Fig. 4C). It remains to be empirically determined whether these ctenophore homologs are active in cleaving collagen/procollagen but assuming they are, breakdown of the ECM during and after wound healing may be a key step in regeneration in *M. leidyi*. Much like other highly regenerative animals, active downregulation of ECM components may be important in *M. leidyi* regeneration. In the GO analysis, we found structural molecule activity (GO:0005198) containing two collagen genes (COL4A6–ML18175a & COL4A6–ML18176a) was enriched in DEG downregulated 3–6hpb (Fig. 3B, Fig. 3D). These two genes are also upregulated in an early time point (1-3hpb) (Supplementary Data 4). Lowered collagen production and degradation of existing collagen proteins may work together to enable *M. leidyi*'s scarless regeneration. Another potential role of these peptidases is in the activation of transforming growth factor-beta (TGF-beta) signaling ligands. TGF-beta signaling is known to initiate wound healing and regeneration in other animals, including axolotls[91–93], but in some contexts can also promote scar formation[94,95]. One of the two TGF-beta ligand genes that have been previously identified in *M. leidyi* (MLTGFbA–ML102235a)[96], is upregulated 3–6hpb. Thus, there may be a functional connection of ECM regulation between peptidase and TGF-beta expression profiles (Supplementary Data 4).

Although not highlighted in the GO analysis, several serine peptidases are differentially expressed. The trypsin-like serine protease gene, PRSS22 (ML11643a) was downregulated 1hpb–3hpb while the transmembrane serine protease TMPRSS3 (ML00576a) was upregulated 3hpb–6hpb (Supplementary Data 4). In the tunicate species *Botrylloides leachi*, inhibition of trypsin-like serine proteases leads to abnormal regenerative outcomes[97]. Trypsin-like serine protease genes were also detected in the gut of the regenerating planarian species *Dugesia japonica and* showed increased expression following bacterial exposure, indicating that they are likely responsive to pathogens, thus integral to innate immunity[98]. A tradeoff between immune function and regeneration has been proposed in other animals from the observation that *Xenopus* larvae have regenerative

capacity but a much weaker immune system[99], and fetal mammals exhibit the potential for scarless wound healing that decreases over time while the immune system is strengthening[100]. Innate immunity in ctenophores is not well understood. However, there is some evidence that pathogenic exposure alters gene expression in *M. leidyi*[101–103]. Whether the chitin-binding genes upregulated by 10mpb play a role in ctenophore immune function is unknown, but recognition and degradation of chitin is widespread in immunity and defense across eukaryotes, especially in marine environments[104]. It is unclear if immunity changes throughout the life of *M. leidyi*, and it has yet to be determined how innate immunity is related to regeneration. Alternatively, the surgery performed in this study could influence the upregulation of these serine peptidases, as it likely exposes the animal to external pathogens and splits the pharynx, which directly affects digestive cells. Finally, caspases are peptidases known to play a key role in apoptosis and, at lower levels, have been suggested to modify cell fate via p53 activity[105,106]. In *M. leidyi*, CASP3 (ML154125a) is downregulated 3–6hpb, suggesting that the regulation of apoptosis may be an important step in regeneration, as shown in many other animals[107–112], or it could be priming the DNA landscape for alterations prior to the onset of cell proliferation at 6 hpb[113], or both (Fig. 4C).

## Biphasic alpha-tubulin gene expression indicates cytoskeletal modifications occur between the middle and late phases of regeneration

The alpha-tubulin genes highlighted by the GO analysis all show late upregulation. However, several also show upregulation in the middle phase of the timeline, the first from 1 to 3 hpb and the second from 24 to 48 hpb. The downregulation that occurs between these two phases during 3–6 hpb, after wound closure but before cell proliferation, is particularly interesting. However, tubulin's diverse functions in cell division, motility, and transport mean that mRNA expression levels alone are insufficient to draw functional conclusions. The simplest explanation for the much later upregulation (24–48 hpb) is that in our bisections, we remove 4 comb rows, and comb rows are rich in tubulin. The *M. leidyi* protein models BLAST reference contains 18 genes annotated as alpha tubulins (Supplementary Data 3). When the median expression across biological replicates of each of these genes is visualized together, it is unclear if any of them show overlapping levels of expression (Supplementary Fig. 4). However, the three alpha tubulin genes included in the consensus DEG as well as the GO enrichment analysis are grouped together in cluster #7, showing that there are substantial similarities in their dynamic patterns of expression (Supplementary Fig. 3H).

## Early response genes are potential initiators of an ancient injury-responsive gene regulation mechanism

We hypothesize that the early response genes identified in *M. leidyi* closely related to early response genes in non-ctenophores represent a core set of injury response components that were present in the last common ancestor of all animals. Although regeneration itself could be generally labeled as inherited or adaptive, individual phases of the process could have developed as a result of either evolutionary trend[114]. As such, the universality of wound healing, along with its position upstream of regeneration, makes the immediate injury response an ideal springboard for comparative analysis across evolutionary history. If there is a core set of ancestral genes involved in the initiation of regeneration, they are likely to be expressed in the earlier phases rather than in species-specific differentiation and morphogenic processes. Robust comparative analysis of gene regulation will help determine if regulatory connections linking the immediate injury response to regeneration-specific genes are highly conserved, ultimately advancing our understanding of cellular control and selective drivers of diverse regenerative outcomes across all animals.

The early response transcription genes identified in *M. leidyi* have been proposed to form part of an early gene regulatory network essential to regeneration across animals, whereby a MAPK signal initiates the expression of bZIP transcription factors which then activate a downstream gene

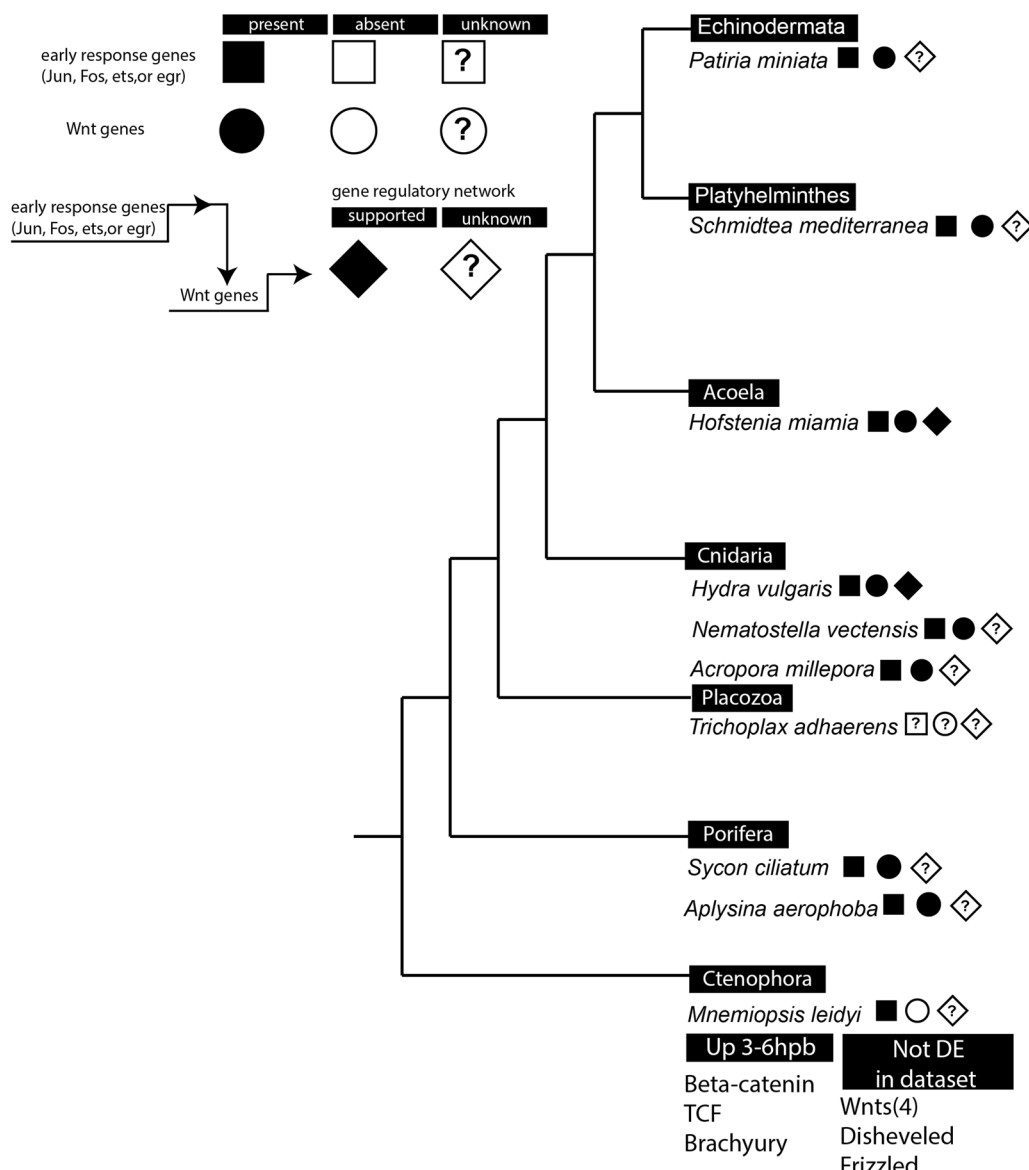

**Fig. 6 | Prevalence of early response and Wnt ligand gene expression across whole-body regenerators.** Up 3–6hpb = differentially upregulated between the 3hpb and 6hpb time points. Not DE in dataset = genes were not recovered as differentially expressed (DE) between any subsequent time points in the entire dataset.

regulatory network, particularly through expression of Wnt ligands[5]. Wnt signaling is a hallmark of regeneration in many animals, particularly in the reestablishment of body axes and the onset of cell proliferation[15,115–120]. Regulatory connections between Fos/Jun expression and the Wnt pathway have been established in a few species (Fig. 6). In *H. vulgaris*, in silico promoter analysis revealed that Wnt ligand genes contain binding sites for bZIP transcription factors[15,73]. In *Drosophila*, AP-1 is also quickly and temporarily expressed proximal to the wound site[50] and the wound-responsive enhancer BRV-B that mediates Wnt signaling contains an AP-1 (i.e., bZIP) binding motif[45]. It is possible that the Wnt pathway contributes to the downregulation of Fos/Jun once wound healing is completed, as it was shown that the inhibitor iCRT14, which blocks the binding of TCF to beta-catenin, also results in prolonged bZIP TF expression in *H. vulgaris*, although this negative regulatory loop has not been functionally established[15]. The prevalence of injury-responsive expression of early response genes (e.g., Jun, Fos, egr, or ets) as well as Wnt ligand genes[4,14,15,44,64,73,84,111,121,122] suggests that these components could be the remnants of an ancient regulatory response for whole body regeneration (Fig. 6). Interestingly, the *M. leidyi* genome contains 4 Wnt ligands, none of which we recovered as DEG[123,124]. However, ML_Beta-catenin, ML_TCF

and the best reciprocal BLAST for Brachyury (TBXT–ML174736a) were identified as upregulated 3–6hpb (Supplementary Data 4). It has been determined that Brachyury can be targeted by Wnt signaling in other systems, but it is still unknown if there is any reliance on Wnt ligand formation for proper regrowth in *M. leidyi*[125]. The lack of differential upregulation of Wnt ligand genes suggests that ctenophore regeneration may differ substantially from other animal groups. Alternatively, while early response genes are widely conserved, they may be regulating the downstream gene expression of other ligands in *M. leidyi*.

There is a growing body of work demonstrating that across species capable of whole-body regeneration, there is a generic wound response that occurs irrespective of the wound context[15,64,73,84,126]. This response includes both early response transcription factors and peptidases, suggesting that they could be the remnants of an ancestral injury response program[69]. Although not specific to wounds that require downstream regeneration versus those that simply heal, components of the generic injury response have been shown to be essential for proper regenerative function. In planarians, it has been shown that the "missing tissue response" specific to regeneration (which includes localized cell proliferation, gene expression changes extending to 24 h post injury, and widespread apoptosis) is not

required, but gene expression changes associated with the generic wound response is essential for proper regeneration[64,127]. In acoels, expression of early response gene egr is required for regenerative success[4], suggesting that the immediate response to injury brings about regulatory changes that could be fundamental to the success of regeneration. Moreover, if generic wound responses are essential for regenerative function, it remains unknown at which point after epithelial closure, the fate of the wound diverges into healing or regeneration.

In *M. leidyi*, localized cell proliferation around the site of injury is dispensable for wound healing, allowing healing to be temporally separable from regeneration[36]. In contrast to planarians, the onset of localized cell proliferation and genes involved in its initiation would be entirely regeneration-specific. Therefore, *M. leidyi* is an advantageous system for dissecting the wound response from the onset of regeneration. The transcriptomic data from our analysis provide a testable hypothesis for gene regulation in response to injury by *M. leidyi*. The candidate early response genes we have identified are generally not highly expressed during development[123] making them viable for gene knockdown methods that have been established in this system[128,129]. Now that we have uncovered the molecular changes underlying regeneration in the last common ancestor, it is possible to construct the GRN underlying regeneration in ctenophores and contribute to the effort to determine the functional basis for the onset of regenerative properties across animals.

## Materials and methods

### Animal collection and husbandry

Adult *Mnemiopsis leidyi* were collected off docks in the Matanzas River estuarine system near St. Augustine and Beverly Beach, FL, and transported to the Whitney Laboratory. Adult animals were kept in a pseudo-Kreisel with open flow-through of filtered, local seawater under constant light and fed enriched artemia several times per day[35]. Prior to spawning, individual animals were isolated in glass finger bowls with freshly filtered seawater and placed in total darkness[130]. After 3 h in the dark, embryos were collected and separated from the adults into fresh bowls filled with full-strength, UV-sterilized, 0.2 µm filtered seawater (FSW). Growing cydippids were fed rotifers (L-type, Reed Mariculture, Inc.) once a day for two weeks[33,35]. Once animals reached 1.5–3 mm in diameter, they were transferred into fresh bowls and subsequently starved for >24 h before surgery.

### Surgeries

Small adults, i.e., cydippids ~2–3 weeks old, of ~1–3 mm body diameter, were used because they are small enough to whole-mount on glass slides, large enough to cut easily, and their surface-to-volume ratio permits easy RNA extraction (which can be inhibited by large amounts of acellular mesoglea relative to cell number in larger animals). For each time point, 20–35 animals were transferred into a 35 mm plastic petri dish coated with silicone (SLYGARD-184, Dow Chemicals, Inc.) to a depth of 2 mm. Using a hand-pulled glass capillary needle; animals were bisected individually within incisions made along the oral-aboral axis at a slightly oblique angle. Halves that included the intact aboral organ along with half the complement of other structures (i.e., 4 comb rows and one tentacle bulb) were retained for extraction. The rest of the animal (the bisected side without the apical organ) was not utilized in these experiments, as it was previously shown they have a lower rate of complete regeneration which might compromise the signal-to-noise ratio in our experiment[24]. For controls, RNA was extracted from animals that did not undergo any surgical procedures but were from the same biological replicate. Biological replicates were each spawned from a unique pool of adults.

### RNA extraction

Using a ThermoFisher (Inc.) RNAqueous Micro Total RNA Isolation kit (AM1912), animals were placed in a lysis buffer, rapidly homogenized by pulse-vortexing, and the cell lysates were subsequently flash frozen in an ethanol/dry ice slurry. Each sample was stored at −80 °C until all time points from each biological replicate were collected. Total RNA was then extracted

according to the kit's manufacturer instructions, and samples were then purified using the Qiagen RNeasy MinElute Cleanup Kit (#74204).

### Library preparation and RNA-seq pipeline

The University of Florida's Interdisciplinary Center for Biotechnology Research Gene Expression and Genotyping Core facility (RRID:SCR_019145) provided standard library preparation and quality assay services (i.e., TapeStation analysis of RNA, library preparation, library quality check, and pooling). Stranded Illumina RNA-Seq libraries were prepared using poly-A selected mRNA with 250 ng as input with 9 amplification cycles. All RNA samples used as library input had RIN > 7. Libraries were sequenced with paired-end, 100 bp reads using the Illumina NovaSeq 6000. The raw reads were transferred and checked for their quality with FastQC v0.11.5. FastQC flagged 0 reads as low quality for all samples (Supplementary Data 1). Reads were then quality filtered and trimmed with Trimmomatic v0.39 (parameters: LEADING:3 TRAILING:3 SLIDINGWINDOW:4:15 MINLEN:36) (https://github.com/dorograce/CydRegenSeq, Supplementary Data 1). Trimmed reads were mapped using Bowtie v2.3.5.1[131] to *Mnemiopsis leidyi* annotated gene models[20] (total number of reads: 1,100,603,638, average: 52,409,697, range: 42,744,538–62,298,240) with all 24 libraries mapping with ≳ 50% alignment rate (average mapping rate: 50.73%, range: 49.69–52.80%). Gene expression was quantified across each library with RSEM[132] (v1.2.28) using default parameters. Count data was exported from RSEM, and all subsequent analysis was performed in R (v4.3.1) using the RStudio IDE[133].

### Differential gene expression analysis

For the NOISeq[37] (v2.44.0) analysis, gene-level counts were input as reads mapped per gene in transcripts per million (TPM), generated from the RSEM quantification[132] (Supplementary Data 2). The NOISeqbio function was integrated to test for differential expression between subsequent time points ($r = 50$), resulting in 834 unique DEG (Fig. 3)[37]. The results from NOISeq show a sequential increase in the total number of DEG (both upregulated and downregulated) starting at the earliest time interval (i.e., uncut–10mpb) and peaking at 3hpb–6hpb (Supplementary Fig. 2A).

Next, differential gene expression analysis was done with edgeR[54] (v3.42.4) on pairs of adjacent time points with a p-value cutoff of 0.05 and p-value adjustment using the Benjamini–Hochberg method. Gene counts were input from the 'expected count' generated from the RSEM quantification step[132], which identified 348 unique genes differentially expressed between adjacent time points, with the greatest number of DEG identified during 3hpb-6hpb (Supplementary Data 2, Supplementary Fig. 2B). An additional analysis using the EBSeq-hmm[55] (v1.34.0) package was performed. Again, TPM gene counts from RSEM were used as input and resulted in a total of 1777 DEG sorted into 161 temporal patterns. To reduce false positives[38] and narrow our candidates for empirical validation, a consensus list of genes identified as DEG by all three analyses was generated (Supplementary Fig. 2D, Supplementary Fig. 3). Using the *M. leidyi* protein models against the NCBI nonredundant human protein database (2020), BLAST protein annotations were assigned to 72 out of the 118 consensus DEG.

### Hierarchical clustering

Hierarchical clustering in the heatmap.2 function in R was used to cluster the temporal gene expression profiles of the genes that comprised our consensus DEG set based on Pearson correlation coefficients[134] with complete linkage (Supplementary Fig. 3).

### Gene ontology

Using the InterPro GO annotations assigned to the *Mnemiopsis* gene models[135], the upregulated and downregulated DEG identified in each interval were tested individually for gene ontology (GO) enrichment using TOPGO[136] (v2.52.0) with the 'classicFisher' significance testing option, with a p-value cutoff of 0.05. REVIGO[137] (v1.12.2) was used to reduce the redundancy of the significant GO terms (threshold = 0.7). Horizontal bar

graphs of the resulting redundancy reduced GO term lists using, with the respective *p*-value of each colored according to each specified gradient (Fig. 3). Individual DEG associated with GO terms were assigned BLAST annotations using the reference generated from the *M. leidyi* protein models (Supplementary Data 6).

## Cloning of in situ probe templates

G-block gene fragments were ordered from IDT for ML_Fos3 (ML182032a), ML_Jun (ML1541120a), and ML_Etslx4 (ML282527a) coding sequences, then A-tailed and ligated each into Promega PGEM-T vector (A3600). Recombinant vectors into *E. coli* were transformed, and individual clones were selected for liquid culture and verified by colony PCR. Plasmids were purified using a Thermo Fisher scientific GeneJET Plasmid Miniprep kit (#K0503) and analyzed by Sanger sequencing to verify the orientation of the sequence (Supplementary Data 8). PCR primers specific to the bacterial promoter sequence were used to amplify the probe template from the minipreps, and DNA was purified using the Monarch PCR & DNA cleanup kit (#T1030L). DIG-labeled RNA probes (antisense and control sense) were transcribed using Invitrogen MEGAscript SP6 (#AM1330) or T7 (#AM1334) kit depending on the orientation of the sequence (Supplementary Fig. 8).

## In situ hybridization

Animals were fixed using Rain-X® primary fixation and aldehyde secondary fixation[62], and subsequent RNA in situ hybridization was performed[138]. A complete protocol is hosted at the following address: https://www.whitney.ufl.edu/media/wwwwhitneyufledu/images/files/RNA-in-situ-hybridization-for-Mnemiopsis-leidyi.pdf. Probes were hybridized using 1 ng/µl probe concentrations at 63 °C[138].

## Statistics and reproducibility

For RNA-seq, three biological replicates were included for each time point. For in situ validation, the time course experiment was performed twice, yielding two biological replicates. Biological replicates were generated for RNA-seq and in situ hybridization by pooling animals from separate spawns from different animals.

## Reporting summary

Further information on research design is available in the Nature Portfolio Reporting Summary linked to this article.

## Data availability

RNA-seq data is available in the NCBI sequencing read archive (SRA) located through this link https://www.ncbi.nlm.nih.gov/bioproject/PRJNA986659. Supplementary data are located and available for download at https://github.com/dorograce/CydRegenSeq/tree/main/05-SUPP_DATA. Additional data is available from the corresponding author upon request.

## Code availability

The code associated with this project is located on our Github page.

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

## Acknowledgements

We thank the Sunset Inlet Homeowners' Association in Beverly Beach and Marker 8 Hotel and Marina in St. Augustine for facilitating animal collections on their properties. We thank Dr. Camille Enjolras, Dr. Fredrik Hugosson, and Dr. Leslie Babonis for their advice in the execution of laboratory protocols. Funding: National Science Foundation Postdoctoral Research Fellowship in Biology under Grant 2010755 (A.E.); National Science Foundation IOS-1755364 (M.Q.M.); National Aeronautics and Space Administration Grants SC37607-01/P0153802 and 80NSSC18K1067 (M.Q.M.).

## Author contributions

D.G.M. performed computational analysis, made probes, and performed surgeries for ISH, curated data, and wrote the paper. A.E. aided in the computational analysis and drafting of the paper. J.R.M. performed surgeries and isolated RNA. J.F.R. aided in computational analysis and data curation. M.Q.M. edited the paper and provided supervision. All authors participated in the conceptualization, experimental design, and revision of the paper and approved the final submission.

## Competing interests

The authors declare no competing interests.
