## [Peer Review File · Communications Biology]

Reviewers' comments:

Reviewer #1 (Remarks to the Author):

In the paper "The highly regenerative ctenophore *Mnemiopsis leidyi* deploys a rapid gene expression response to injury that dates back to the last common animal ancestor" the authors generated bulk RNA sequencing libraries sampling the regeneration of the Ctenophore *Mnemiopsis* up until 48hrs post amputation, by collecting the apical organ containing side of the regenerate at various intervals. From these data, the authors identify three distinct transcriptomic responses over the 48 hour period. The authors equate these periods with an initial wound healing response that peaks at 1hr, followed by an intermediate response prior to cell proliferation (1-6), and finally a regeneration response (6-48hr) wherein the tissue balance is restored. By applying three separate algorithms for estimating differential gene expression, the authors identify 304 genes that were identified across all methods. Of these, approximately 2/3 (196/304) could be assigned any putative identity from their approach of using best-blast hit with annotated human genes. They then proceed to investigate the content of the gene sets in the three phases and identify three principal categories of genes active in each phase: early response transcription factors are activated, followed by activation of peptidases and then downregulation of cytoskeletal genes. The authors also validate their finding via in situ hybridization, demonstrating that the early response gene set is indeed expressed at the wound site. Finally, the authors claim that the early gene set, which included orthologs of known early response wound healing genes from other animals, represents an early transcriptomic program for wound response that is shared by all animals.

The experimental set up provides a wealth of molecular data to start dissecting regeneration in this enigmatic animal model. The analyses of these data are somewhat preliminary and could use additional validation and supporting information. In general, I feel more care is required in describing the observations in the data as presented. At present there are many mis-leading descriptions in the main text with regards to changes in gene expression: in particular I refer to the tubulins, however this casts doubt on all descriptions and so please revisit all data descriptions carefully to ensure that your text indeed reflects what you show as data. I would also like to see some expression plots of some genes that fall outside the set of DEGs (from any of the approaches), perhaps as a supplement. To select genes, other members of the discussed gene families (tubulins, Fos, etc) could be shown to demonstrate that the DEG methods accurately select responsive genes, and the other paralogs show a random and/or consistent expression profile across the experiment. In addition, some re-organization of the text to enhance clarity of the main message(s) is required before I can recommend publication. Below I highlight some of these points that came up as I went through the manuscript.

Ln53: please expand upon "the shared wound response genes" – which genes have already been identified in this category?

Ln70-73: I assume this statement is included because it is the younger stage that is investigated? Please clarify. What stage is used in this work and why was it chosen?

Ln77: "op. cit." ... ?

Ln120-125: I am not familiar with this algorithm and find this description of "expression path" confusing and do not see how the stated pattern then indicates "upregulation during 1.3 hpb and downregulation from 3-6 hpb". Please clarify this for the reader.

Ln125-127: Please state clearly how many DEGs were looked at in this way, and how many of these have putative identities assigned based on this method. A very limited subset of gene models were identified in this way (2/3 of the consensus set according to the methods section: and what about the rest of the gene models?). Please address the remaining genes somewhere in the article. I would suggest that using a broader annotation strategy could allow you to assign putative identities to a larger set of the genes. If not, then are these all Ctenophore-specific genes? In this case, what might this imply about your results?

Ln131: Please explain in the main text why you choose to show the NOISEq results here and not the

other analyses? What makes this analysis worthy of greater visualization here whereas the other two are in the supplement (and refer to where in the supplement to find the others for readers who may be interested). I would also suggest that the full output of identified genes for each method should accompany the paper with the corresponding method statistics.

Fig.3: It is unclear to me what the coloration indicates. Why are only some of the up(down)regulated genes highlighted? What makes these special compared to the rest of the genes listed on the figure? This also applies to the similar figure in the supplement (sup. Fig.1).

Ln 144: Please specify here how many genes are in the consensus list (and how many are identified – commented above). Otherwise the reader is forced to hunt through the manuscript until the methods section to find this information. This is relevant to assessing the conclusions and so should be stated clearly upfront.

Ln146: Please provide more information regarding the gene clusters – give the full content of the clusters in some format including how many genes are in each, how many may be composed of uninformative gene models (ie. No Human reciprocal best blast hit). I would also suggest that the expression profiles in supplementary figure 3 are worthy of being in the main document.

Supplementary Figure 3: Are the genes shown in the clustered order? If not, please do so and indicate the clusters on the figure.

Ln161: It confuses me why term 4 is combined here with 2/5. Figure 4B clearly shows overlap / similarity of terms 2/5 but not 4. Why are these considered together here? How is scavenger receptors indicative of extracellular matrix degradation?

Ln181: "surrounding" implies before AND after. Change to "before" or similar.

Ln190: remove "gene" and "its" (2x) from the sentence. Here you take genes from each GO category and plot their expression profiles but each GO set that you look at responds differently. This should be reflected in the title of the section.

Ln 194-200: Regarding 'scavenger receptors': here you identify two member genes that may have peptidase activity and state that these are upregulated in the later response window. PRSS22 however appears to be downregulated between 1-3 hr and then return to pre-amputation levels.

Ln217: Please plot also the other 14 tubulins for comparison. The tubulins as shown also indicate downregulation after 1hr then return to control levels. In the text this is described incorrectly compared to the control "increased expression 1-3 hpb" – the data as illustrated shows relatively stable levels compared to the control and then downregulation (between 1:6 hr although the 3hr timepoint is at control levels for all genes) with gradual increase back to control levels again by 48hr. Please revise the text to accurately reflect the observation in the data.

Fig.5: To aid in interpretation of this figure, I would suggest that the key (lower right corner) be presented first (upper left corner). Which, if any, of the expression differences show statistically significant differences? In some cases, there are no clear patterns (i.e. A and B: each gene is behaving differently).

Ln235: is there evidence in the literature that a MAP kinase cascade performs this role in other systems? Please cite this here... otherwise, what is the rationale for suggesting this here?

Ln240-246: As written here, you state that secondary response genes are activated by the early response genes. Thus one would expect these to be up-regulated with a delay, such as you see between the 10min responders vs. the 1hr responders. However you then identify genes that show a clear downregulation (and thus are repressed, not activated) in the 3hr sample as reasonable secondary response candidates. This logic in my opinion is flawed and an alternative interpretation of the role of these genes is necessary.

Ln260: Here the gene clustering is re-introduced (first reference at Ln146, see above comment). Please provide more information regarding the clustering: how many genes per cluster, which genes in each cluster.

Ln282: mis-placed period.

Ln283-284: This can not be stated based on the taxonomic distribution used in the trees. What is Supplementary file X?

Fig6: controls are in supplement 6/7 not 5/6.

Discussion: The first three paragraphs are incomplete and disorganized:

Ln332: section title unclear. I think the point of this section is that the upregulation of various types of peptidases could explain the absence of scarring, which is shown to be detrimental to regeneration in other systems?

This section needs some re-organization and streamlining in order to convey this point. Please try to avoid three sentence paragraphs (first two paragraphs presented).

Ln399: is there any evidence for other apoptosis related genes up-regulated at this stage? You have the entire transcriptome available and so there is no reason that you cannot look specifically at other genes even if they are not on the 'consensus DEG' list. This list is a great starting point to dig into and start making sense of the data, but you should then attempt to test hypotheses generated by looking for additional evidence: in this case you hypothesize that apoptosis may play a role in the wound healing response. Do other apoptosis-related genes support this? Could you look into TUNEL staining to provide additional evidence for this phenomena?

Ln406: This section needs re-writing. There is no upregulation documented for any of the tubulins that is presented in the figures. Only downregulation followed by re-establishment of control levels. This more likely represents a global shutdown of transcription prior to re-entry into the cell cycle, however this hypothesis would need to be tested with the data for supporting evidence that many 'housekeeping / structural' type genes are repressed at this transition.

Ln430: Two sentences are insufficient to make a paragraph.

Ln432: "are activated by the immediate injury response" – if they are upregulated within minutes is this not the immediate injury response? What do you refer to as the "immediate injury response"?

Ln467: I think the wnt response is also a very interesting observation and supp. Figure 8 merits being part of the main paper.

Ln469: You show that wnt ligands are upregulated at the 6h point, which corresponds to the onset of cell cycling. Thus is it not more parsimonious to suggest that these ligands could be activated by the early response genes rather than as you suggest at the end of this paragraph that the early response bZIP genes are regulated by wnt?

Ln491: This discussion of ROS is out of context of the current work. This does not need to be included here as an entire paragraph.

Ln 532: please define what how you distinguish "healing" and "regeneration".

Ln 549: Please use passive voice for the methods: "Adult M. leidyi were collected" rather than "we collected". This becomes especially relevant for the section starting Ln580 where it becomes "they...". If the production of the data is so heavily dependent on "them", then "they" should be co-authors and thus is becomes "we". Using the passive voice to describe the methods would alleviate this problem.

Ln567: What happened to the other half then? Can they still regenerate without the AO? Please justify / explain this strategy / choice in experimental design.

Ln616: Where is Supplementary File X?

Ln623: please show the clustering in the figure. See also previous comments to this regard.

Ln676: Please review your reference list. There is an issue with your reference manager and some items are improperly formatted. For example: n104, n106; others are missing essential information. Some last additional thoughts:

The main conclusion that jun/fos are immediate early genes that are activated by injury in Ctenophores is an important (albeit not necessarily surprising) observation. Perhaps if the paper included more focus on the uncertain phylogenetic position of this model, and highlight some of the most recent work suggesting that Ctenophores are very old and perhaps have some interesting convergences (Schultz et al 2023 (<https://doi.org/10.1038/s41586-023-05936-6>) genome organization suggesting Ctenophores are basal-most on the metazoan tree, and Burkhardt et al 2023 (DOI: 10.1126/science.ade564) on the syncytial nervous system), then this observation would hold more interest for a broader audience. The fact that this may also involve wnt signalling is also very interesting in this context, but not adequately presented in the current manuscript. This could be

explored further in the results perhaps rather than as an afterthought at the end of the discussion. Although I recommend some major revisions to the work, I think the dataset generated here is well thought out and will serve as a valuable resource for exploration of the molecular side of regeneration in this system. As such, I feel it important that the observations made of the data are carefully and fully presented, including full gene lists and annotation information from the various informatic approaches applied to the data.

Reviewer #2 (Remarks to the Author):

This is an interesting ms on regeneration in the ctenophore *Mnemiopsis leidyi*. There are as yet no comprehensive molecular regeneration studies on ctenophores, a group that has recently gained importance through studies on the synteny of chromosomal organization of genes in metazoans as the oldest recent phylogenetic form of metazoans. Studies on these animals will undoubtedly shed light on the conservation of basic regenerative mechanisms throughout the animal kingdom. While there is evidence from other basal forms (cnidarians, flatworms) that injury signals are conserved, we do not yet understand how these responses are translated into patterning signals. On this level, it is quite possible that lineage-specific mechanisms do exist. To address these questions, the authors performed regeneration experiments in *Mnemiopsis* by cutting animals longitudinally along their oral-aboral axis and sampling the regenerating halves at different time points (0-96 hours). At early stages of regeneration, they found a transient upregulation of injury-specific genes (encoding peptidases, cytoskeletal and transcription factors) at the site of wounding. Previous work has shown that regeneration in *Mnemiopsis* involves wound healing (2 hours), cell proliferation (from 6 hours) and new formation of all missing structures/cell types (up to 48 hours), with no apparent scar or blastema formation. The authors now show that the most active period of differential gene expression occurs between 1 and 6 hours after injury. Gene ontology analysis of the consensus DEG list revealed that peptidases, transcription factors, and elements of the cytoskeleton were significantly increased. These are important data that are of interest to a broad readership. However, there are several points that need to be addressed.

This work is largely descriptive. Expression profiling shows that transcription factors (bZIP and ETS motifs) as well as genes encoding for members of the Wnt signaling pathway are activated before the onset of cell proliferation. These data coincide in many respects with what is already known about the molecular processes in early regeneration or a response to injury signals in several well-studied basal model organisms (cnidarians, flatworms). The data certainly allow the hypothesis that an "ancient gene regulation network" is acting in early regeneration. However, the data are purely descriptive and their functional statements are based exclusively on findings that have been experimentally elaborated in other basal metazoans, especially the cnidarians and flatworms.

The entire functional context gets established in the discussion, making this interesting to read, but in the end it is based on findings in other organisms like the discussion on early response genes (*Fos3*, *Jun*, peptidase genes) and TGF-beta and Wnt signaling. made in the summary figure are plausible, but without any experimental validation. It would also be desirable if the work included proteomic data that could give an indication of the triggering of the injury signal. Data on redox, Ca²⁺ and MAPK signals are also missing.

In the current version of the discussion, the mechanism of regeneration in ctenophores is postulated only from comparison with other organisms and the expression profiles. However, these data only gain weight from the fact that ctenophores are probably the most basal recent animal group. One solution to highlight the relevance of the data presented here to a wider readership would be, for example, a

tabular presentation of selected early and late genes of regeneration in ctenophores in a comparison with cnidarians, sponges and selected bilaterians, such as the flatworms. This could also be accompanied by a summary of the text of the discussion.

Reviewer #3 (Remarks to the Author):

This manuscript describes transcriptional responses across wound healing and regeneration in the ctenophore *Mnemiopsis*. Because ctenophores are the sister group to all other animals, information from this group is critical for reconstructing the ancestral injury response of animals. Using RNA-Seq, this study identifies several suites of genes that are strongly differentially expressed across the regeneration timecourse. The authors find that a number of genes known to be involved in early wound healing and the transition to regeneration in other animals are differentially expressed in a consistent manner in this ctenophore, and in situ support their involvement in the wound response. The authors conclude that the early molecular response to injury, including the transition from wound healing to regeneration initiation, has broadly conserved components that date back to the common ancestor of animals. They use their results to build a plausible ancestral gene regulatory network for the initiation of animal regeneration.

The data presented are clear and convincing and the interpretation is well contextualized and justified. The findings are of high importance for understanding animal regeneration as they reveal ancient conserved molecular responses to injury and allow the authors to generate testable hypotheses regarding injury induced gene networks. This work represents a very significant advance for the field of animal regeneration. The manuscript is well written. I have only one general comment and a number of smaller editorial comments.

General comment:

1. The authors choose to focus nearly exclusively on DEGs that suggest conserved roles in regeneration. It would be useful to describe, at least briefly, some of the components of the injury response that are found in other animals but that were not recovered as strong DEGs in the ctenophore dataset (and possibly some of the strong DEGs that have been found in this study but not previously implicated in regeneration). This will be important for putting the "conservation", that is the emphasis of the paper, into context, and will also suggest how regeneration pathways have evolved.

Specific comments:

2. Fig 4B – Not convinced this figure is needed. If kept, figure legend should include the names of clusters 6-10 since they are numbered in the figure.
3. 117 – need close parentheses
4. Fig 6 legend – extra parenthesis, aboral misspelled
5. 257 – edit – missing/wrong word
6. 257 – indicate Supplemental file number
7. 341-343 – edit for grammar
8. 354-356 – unclear – tradeoffs don't arise from impaired differentiation.
9. 414 – what third upregulation? this was not clearly described prior to this point
10. 463 – prologues – odd word choice
11. 491 – remove "signaling"
12. 506 – regeneration could be both adaptive and inherited. Clarify.

We have included supplementary files listed in the table below

File Name	Description
Supplementary File 1	FastQC of 1. Raw reads 2. Trimmed reads
Supplementary File 2	Quantification of genes from RSEM
Supplementary File 3	Reciprocal best BLAST hit for each M. leidyi gene model and corresponding e-value
Supplementary File 4	Lists of DEG from each method, used to generate the consensus DEG
Supplementary File 5	DEG Lists with p-values/log2FC values 1. EdgeR 2. NOISeq 3. EBSeqHmm as shown in Supplementary Figure 2
Supplementary File 6	Enriched terms for each time interval and each direction (Up or Down). Under each interval contains the ML gene associated with that term and its corresponding reciprocal best BLAST hit
Supplementary File 7	Tree files associated with the BZIP and ETS protein family trees (see Supplementary Figure 5 and 6)
Supplementary File 8	RNA probe sequences aligned to ML gene models

NLn = New line in revised manuscript

In particular, please note that the following revisions would be necessary for us to contact our referees again:

The authors choose to focus nearly exclusively on DEGs that suggest conserved roles in regeneration. It would be useful to describe, at least briefly, some of the components of the injury response that are found in other animals but that were not recovered as strong DEGs in the ctenophore dataset

We have now highlighted that a striking majority of studies of gene expression during whole body regeneration have discovered Wnt signaling genes. Although we did find several genes associated with Wnt signaling upregulated during the timeline, we did not recover any Wnt ligand genes. This is in contrast to what has been found in other systems and provides an interesting question for future research (See figure 6)

Ln120-125: I am not familiar with this algorithm and find this description of "expression path" confusing and do not see how the stated pattern then indicates "upregulation during 1.3 hpb and downregulation from 3-6 hpb". Please clarify this for the reader.

NLn 262-264: We clarified that the results listed as "expression path" from EBseq-hmm (up-up-down,etc) correspond to the ordered time intervals (Uncut-10mpb,10mpb-

1hpb,etc). Therefore, since the greatest number of genes were categorized into the expression path "Up-Down-Down-Up-Down-Down-Up", that shows that they were "Up" in the fourth position (during 3hpb-6hpb).

Ln125-127: Please state clearly how many DEGs were looked at in this way, and how many of these have putative identities assigned based on this method

NLn 143-144: We performed a reciprocal BLAST on all of the *Mnemiopsis* protein models to the NCBI nonredundant human protein database (2020). The identifiers (i.e. ML0000x) are the same for the protein and gene models, therefore annotating the entire population of protein models with human protein annotations allowed us to generate a reference for annotating the genes identified as DEG in each method.

NLn 266-267: For the 118 genes in the consensus, we subset the list of total gene models and respective best BLAST hit.

Ln131: Please explain in the main text why you choose to show the NOISeq results here and not the other analyses.

NLn: 122-125. We clarified our approach in this portion the new main text, stating that we used the data from NOISeq to describe each phase of the regenerative process with respect to each time interval because this method is more suitable for this experimental set up (3 biological replicates, 7 timepoints, see <https://doi.org/10.1093/bib/bbx115>). We chose to add a Gene ontology analysis for each gene list generated from NOISeq and annotate the corresponding box and whisker plots with the intervals in which they are DEG. The individual results for the three methods are included in Supplementary Figure 2 and were used to generate the consensus list that narrowed down or search for candidate genes.

Fig.3: It is unclear to me what the coloration indicates.

We changed the key to "Direction of expression" to show that the quantity of genes in each time interval that are upregulated are represented by yellow in the bar graph and those that are downregulated are blue. We have moved this figure to include all three methods together in the Supplement (Supplementary Figure 2).

Ln 144: Please specify here how many genes are in the consensus list

NLn 264: We have now included this

Ln146: Please provide more information regarding the gene clusters

NLn 269: We have now clarified that we clustered the genes in the consensus to identify high confidence early response genes. Each gene cluster can now be found in Supplementary Figure 3.

Ln 194-200: Regarding 'scavenger receptors': here you identify two member genes that may have peptidase activity and state that these are upregulated in the

later response window. PRSS22 however appears to be downregulated between 1-3 hr and then return to pre-amputation levels.

NLn 430: We have removed this erroneous gene expression observation and instead included that this gene is labeled as downregulated 1-3hpb.

Ln217: Please plot also the other 14 tubulins for comparison. Please revise the text to accurately reflect the observation in the data.

NLn 467-471: We now include a multi-line graph for all 18 alpha tubulin genes has been generated and can be found in Supplementary Figure 4.

Ln235: is there evidence in the literature that a MAP kinase cascade performs this role in other systems? Please cite this here... otherwise, what is the rational for suggesting this here?

NLn 192: We now include this citation

Ln260: Here the gene clustering is re-introduced (first reference at Ln146, see above comment). Please provide more information regarding the clustering: how many genes per cluster, which genes in each cluster.

More detail has been added to the text about the methods for hierarchical clustering from the heatmap as well as an updated figure (Supplementary Figure 3) showing all of the clusters

Ln282: mis-placed period.

NLn 292: This is now removed

Ln283-284: This can not be stated based on the taxonomic distribution used in the trees. What is Supplementary file X?

We have removed the erroneous duplication inference. Supplementary File type is now fixed.

Fig6: controls are in supplement 6/7 not 5/6.

This is now fixed in the figure legend. They are now included in Supplementary Figure 7 & 8, respectively

Discussion: The first three paragraph are incomplete and disorganized. Pleae restructure according to the reviewer comments

We have now restructured the discussion

Reviewers' comments:

Reviewer #1 (Remarks to the Author):

In the paper "The highly regenerative ctenophore Mnemiopsis leidyi deploys a rapid gene expression response to injury that dates back to the last common animal ancestor" the authors generated bulk RNA sequencing libraries sampling

the regeneration of the Ctenophore *Mnemiopsis* up until 48hrs post amputation, by collecting the apical organ containing side of the regenerate at various intervals. From these data, the authors identify three distinct transcriptomic responses over the 48 hour period. The authors equate these periods with an initial wound healing response that peaks at 1hr, followed by an intermediate response prior to cell proliferation (1-6), and finally a regeneration response (6-48hr) wherein the tissue balance is restored. By applying three separate algorithms for estimating differential gene expression, the authors identify 304 genes that were identified across all methods. Of these, approximately 2/3 (196/304) could be assigned any putative identity from their approach of using best-blast hit with annotated human genes. They then proceed to investigate the content of the gene sets in the three phases and identify three principal categories of genes active in each phase: early response transcription factors are activated, followed by activation of peptidases and then downregulation of cytoskeletal genes. The authors also validate their finding via in situ hybridization, demonstrating that the early response gene set is indeed expressed at the wound site. Finally, the authors claim that the early gene set, which included orthologs of known early response wound healing genes from other animals, represents an early transcriptomic program for wound response that is shared by all animals.

The experimental set up provides a wealth of molecular data to start dissecting regeneration in this enigmatic animal model. The analyses of these data are somewhat preliminary and could use additional validation and supporting information. In general, I feel more care is required in describing the observations in the data as presented. At present there are many mis-leading descriptions in the main text with regards to changes in gene expression: in particular I refer to the tubulins, however this casts doubt on all descriptions and so please revisit all data descriptions carefully to ensure that your text indeed reflects what you show as data. I would also like to see some expression plots of some genes that fall outside the set of DEGs (from any of the approaches), perhaps as a supplement. To select genes, other members of the discussed gene families (tubulins, Fos, etc) could be shown to demonstrate that the DEG methods accurately select responsive genes, and the other paralogs show a random and/or consistent expression profile across the experiment. In addition, some re-organization of the text to enhance clarity of the main message(s) is required before I can recommend publication. Below I highlight some of these points that came up as I went through the manuscript.

Ln53: please expand upon "the shared wound response genes" – which genes have already been identified in this category?

NLn 52-54: The specific genes have now been included

Ln70-73: I assume this statement is included because it is the younger stage that is investigated? Please clarify. What stage is used in this work and why was it chosen?

NLn 76-78: We have now provided a justification for using this stage

Ln77: "op. cit." ... ?

We have removed this

Ln125-127: Please state clearly how many DEGs were looked at in this way, and how many of these have putative identities assigned based on this method. A very limited subset of gene models were identified in this way (2/3 of the consensus set according to the methods section: and what about the rest of the gene models?). Please address the remaining genes somewhere in the article. I would suggest that using a broader annotation strategy could allow you to assign putative identities to a larger set of the genes. If not, then are these all Ctenophore-specific genes? In this case, what might this imply about your results?

NLn 143-146: We have now included the total number of BLAST hits designated to the entire population of gene models. This 'reference' list was used to annotate the ML genes identified in each method individually as well as the consensus. Although 72/118 genes in the consensus were annotated, the clustering of the entire consensus list (Supplementary Figure 3) allowed us to visualize genes that did not include a BLAST annotation. We do not interpret these remaining 42 genes as ctenophore-specific, however, when viewed in the *Mnemiopsis* genome portal, many of them do not have a BLAST hit across the various species listed (e.g., <https://research.nhgri.nih.gov/mnemiopsis/wiki/index.php/ML018013a>) Any of the unannotated ML genes identified in our analysis can be viewed on the genome portal.

Ln131: Please explain in the main text why you choose to show the NOISeq results here and not the other analyses? What makes this analysis worthy of greater visualization here whereas the other two are in the supplement (and refer to where in the supplement to find the others for readers who may be interested). I would also suggest that the full output of identified genes for each method should accompany the paper with the corresponding method statistics.

Please refer the response to this comment in the first section

Fig.3: It is unclear to me what the coloration indicates. Why are only some of the up(down)regulated genes highlighted? What makes these special compared to the rest of the genes listed on the figure? This also applies to the similar figure in the supplement (sup. Fig.1).

This figure has now been moved to the supplement (Supplementary Figure 2). The colored bars are showing the actual number of DEG identified in each direction (up or down) on the y-axis and the names listed in each interval are the top 20 DEG according to p-value or log2FC. There is no significance to the genes are actually overlapping with the bar, if there were 20 genes in that category, the top 20 are included and ordered. However, some categories contained fewer than 20 genes, and many of them were short enough to not extend past the bar.

Ln 144: Please specify here how many genes are in the consensus list (and how many are identified – commented above). Otherwise the reader is forced to hunt through the manuscript until the methods section to find this information. This is relevant to assessing the conclusions and so should be stated clearly upfront. Please refer the response to this comment in the first section

Ln146: Please provide more information regarding the gene clusters – give the full content of the clusters in some format including how many genes are in each, how many may be composed of uninformative gene models (ie. No Human reciprocal best blast hit). I would also suggest that the expression profiles in supplementary figure 3 are worthy of being in the main document. Please refer the response to this comment in the first section

Supplementary Figure 3: Are the genes shown in the clustered order? If not, please do so and indicate the clusters on the figure.
Yes, the genes ordered in each row on the heatmap are ordered according to cluster (see dendrogram in Supplementary Figure 3)

Ln161: It confuses me why term 4 is combined here with 2/5. Figure 4B clearly shows overlap / similarity of terms 2/5 but not 4. Why are these considered together here? How is scavenger receptors indicative of extracellular matrix degradation?

The organization of this figure has changed since the previous submission and our new gene ontology analysis(Figure 4).

Ln181: "surrounding" implies before AND after. Change to "before" or similar.
NLn 215: We have changed the paragraph heading to “Peptidase transcripts are downregulated prior to the onset of cell proliferation”

Ln190: remove "gene" and "its" (2x) from the sentence. Here you take genes from each GO category and plot their expression profiles but each GO set that you look at responds differently. This should be reflected in the title of the section.
NLn 452-454: This has been fixed in the text to accurately describe the regulatory pattern of this gene.

Fig.5: To aid in interpretation of this figure, I would suggest that the key (lower right corner) be presented first (upper left corner). Which, if any, of the expression differences show statistically significant differences? In some cases, there are no clear patterns (i.e. A and B: each gene is behaving differently).

This is now included as Figure 4. We have moved the key to the top right corner. We have now annotated each expression plot with the differential expression calls (Up or down) between subsequent time intervals along the X-axis to aid in our interpretation of patterns.

Ln240-246: As written here, you state that secondary response genes are activated by the early response genes. Thus one would expect these to be up-regulated with a delay, such as you see between the 10min responders vs. the 1hr responders. However, you then identify genes that show a clear downregulation (and thus are repressed, not activated) in the 3hr sample as reasonable secondary response candidates. This logic in my opinion is flawed and an alternative interpretation of the role of these genes is necessary.

NLn 203-205: We have now proposed two genes that are downregulated 1-3hpb and subsequently upregulated 3-6hpb. Although this pattern can be suggestive of a delay, it is upregulated considerably after what we have proposed to be “early response genes”. Now, in the text we shed doubt on the direct regulation of these genes by the “early response genes”

Ln260: Here the gene clustering is re-introduced (first reference at Ln146, see above comment). Please provide more information regarding the clustering: how many genes per cluster, which genes in each cluster.

Please refer the response to this comment in the first section

Ln332: section title unclear. I think the point of this section is that the upregulation of various types of peptidases could explain the absence of scarring, which is shown to be detrimental to regeneration in other systems? This section needs some re-organization and streamlining in order to convey this point. Please try to avoid three sentence paragraphs (first two paragraphs presented).

NLn 387: We have changed the section title to “Peptidase transcripts are highly regulated in the scarless regeneration of *M. leidy*” and reorganized the text.

Ln399: is there any evidence for other apoptosis related genes up-regulated at this stage? You have the entire transcriptome available and so there is no reason that you cannot look specifically at other genes even if they are not on the 'consensus DEG' list. This list is a great starting point to dig into and start making sense of the data, but you should then attempt to test hypotheses generated by looking for additional evidence: in this case you hypothesise that apoptosis may play a role in the wound healing response. Do other apoptosis-related genes support this? Could you look into TUNEL staining to provide additional evidence for this phenomena?

Apoptosis in *M. leidy* is still unknown but could be of particular interest due to their rapid and scarless wound healing. The gene ontology term ‘apoptotic process’ (GO:0006915) was not assigned to any gene in the *M. leidy* Interproscan, therefore it is unclear if there are additional genes that are involved and the Caspase gene appears to be the best candidate for future experimentation.

Ln406: This section needs re-writing. There is no upregulation documented for any of the tubulins that is presented in the figures. Only downregulation followed

by re-establishment of control levels. This more likely represents a global shutdown of transcription prior to re-entry into the cell cycle, however this hypothesis would need to be tested with the data for supporting evidence that many 'housekeeping / structural' type genes are repressed at this transition.

NLn 460: This section has been re-written to accurately reflect the statistically significant changes in gene expression.

Ln430: Two sentences are insufficient to make a paragraph.

NLn 333: this paragraph has been reworked

Ln432: " are activated by the immediate injury response" – if they are upregulated within minutes is this not the immediate injury response? What do you refer to as the "immediate injury response"?

NLn 335: We have changed the working to “participating in the immediate injury response” as we have labeled them as early response genes that make up the immediate response

Ln467: I think the wnt response is also a very interesting observation and supp. Figure 8 merits being part of the main paper.

We have now integrated the discussion of Wnt pathway genes in *M. leidyi* in the discussion text as well as Figure 6

Ln469: You show that wnt ligands are upregulated at the 6h point, which corresponds to the onset of cell cycling. Thus is it not more parsimonious to suggest that these ligands could be activated by the early response genes rather than as you suggest at the end of this paragraph that the early response bZIP genes are regulated by wnt?

NLn 516-519: We have now proposed this in the discussion

Ln491: This discussion of ROS is out of context of the current work. This does not need to be included here as an entire paragraph.

NLn 376-386: We have now broadened this to discuss the possible upstream activation of early response genes

Ln 532: please define what how you distinguish "healing" and "regeneration".

NLn 534: We have included that this divergence likely occurs after epithelial closure

Ln 549: Please use passive voice for the methods: "Adult *M. leidyi* were collected" rather than "we collected". This becomes especially relevant for the section starting Ln580 where it becomes "they...". If the production of the data is so heavily dependent on "them", then "they" should be co-authors and thus is becomes "we". Using the passive voice to describe the methods would alleviate this problem.

All of the methods have now been changed to passive voice

Ln567: What happened to the other half then? Can they still regenerate without the AO? Please justify / explain this strategy / choice in experimental design.

NLn 573-575: We have now added this justification along with reference to the original work that showed this

Ln616: Where is Supplementary File X?

This has been fixed

Ln623: please show the clustering in the figure. See also previous comments to this regard.

Clustering can now be seen in Supplementary Figure 3

Ln676: Please review your reference list. There is an issue with your reference manager and some items are improperly formatted. For example: n104, n106; others are missing essential information.

This has been fixed

Reviewer #2 (Remarks to the Author):

This is an interesting ms on regeneration in the ctenophore *Mnemiopsis leidyi*. There are as yet no comprehensive molecular regeneration studies on ctenophores, a group that has recently gained importance through studies on the synteny of chromosomal organization of genes in metazoans as the oldest recent phylogenetic form of metazoans. Studies on these animals will undoubtedly shed light on the conservation of basic regenerative mechanisms throughout the animal kingdom. While there is evidence from other basal forms (cnidarians, flatworms) that injury signals are conserved, we do not yet understand how these responses are translated into patterning signals. On this level, it is quite possible that lineage-specific mechanisms do exist. To address these questions, the authors performed regeneration experiments in *Mnemiopsis* by cutting animals longitudinally along their oral-aboral axis and sampling the regenerating halves at different time points (0-96 hours). At early stages of regeneration, they found a transient upregulation of injury-specific genes (encoding peptidases, cytoskeletal and transcription factors) at the site of wounding. Previous work has shown that regeneration in *Mnemiopsis* involves wound healing (2 hours), cell proliferation (from 6 hours) and new formation of all missing structures/cell types (up to 48 hours), with no apparent scar or blastema formation. The authors now show that the most active period of differential gene expression occurs between 1 and 6 hours after injury. Gene ontology analysis of the consensus DEG list revealed that peptidases, transcription factors, and elements of the cytoskeleton were significantly increased. These are important data that are of interest to a broad readership. However, there are several points that need to be addressed.

This work is largely descriptive. Expression profiling shows that transcription factors (bZIP and ETS motifs) as well as genes encoding for members of the Wnt signaling pathway are activated before the onset of cell proliferation. These data coincide in many respects with what is already known about the molecular processes in early regeneration or a response to injury signals in several well-studied basal model organisms (cnidarians, flatworms). The data certainly allow the hypothesis that an "ancient gene regulation network" is acting in early regeneration. However, the data are purely descriptive and their functional statements are based exclusively on findings that have been experimentally elaborated in other basal metazoans, especially the cnidarians and flatworms.

The entire functional context gets established in the discussion, making this interesting to read, but in the end it is based on findings in other organisms like the discussion on early response genes (Fos3, Jun, peptidase genes) and TGF-beta and Wnt signaling. made in the summary figure are plausible, but without any experimental validation. It would also be desirable if the work included proteomic data that could give an indication of the triggering of the injury signal. Data on redox, Ca²⁺ and MAPK signals are also missing.

In the current version of the discussion, the mechanism of regeneration in ctenophores is postulated only from comparison with other organisms and the expression profiles. However, these data only gain weight from the fact that ctenophores are probably the most basal recent animal group. One solution to highlight the relevance of the data presented here to a wider readership would be, for example, a tabular presentation of selected early and late genes of regeneration in ctenophores in a comparison with cnidarians, sponges and selected bilaterians, such as the flatworms. This could also be accompanied by a summary of the text of the discussion.

With the addition of Figure 3 we hope to provide a more holistic description of gene expression while also highlighting how these changes are distinct for each time interval and regulatory direction. With the addition of Figure 6, we hope to clearly illustrate how highly conserved early response gene expression is across a variety of animal phyla. In addition, we hope to encourage future experimentation to determine the evolutionary significance tied to the absence of wnt ligand expression following injury in *M. leidy*.

Reviewer #3 (Remarks to the Author):

This manuscript describes transcriptional responses across wound healing and regeneration in the ctenophore *Mnemiopsis*. Because ctenophores are the sister group to all other animals, information from this group is critical for reconstructing the ancestral injury response of animals. Using RNA-Seq, this study identifies several suites of genes that are strongly differentially expressed across the regeneration timecourse. The authors find that a number of genes known to be involved in early wound healing and the transition to regeneration in other animals are differentially expressed in a consistent manner in this

ctenophore, and in situ support their involvement in the wound response. The authors conclude that the early molecular response to injury, including the transition from wound healing to regeneration initiation, has broadly conserved components that date back to the common ancestor of animals. They use their results to build a plausible ancestral gene regulatory network for the initiation of animal regeneration.

The data presented are clear and convincing and the interpretation is well contextualized and justified. The findings are of high importance for understanding animal regeneration as they reveal ancient conserved molecular responses to injury and allow the authors to generate testable hypotheses regarding injury induced gene networks. This work represents a very significant advance for the field of animal regeneration. The manuscript is well written. I have only one general comment and a number of smaller editorial comments.

The authors choose to focus nearly exclusively on DEGs that suggest conserved roles in regeneration. It would be useful to describe, at least briefly, some of the components of the injury response that are found in other animals but that were not recovered as strong DEGs in the ctenophore dataset (and possibly some of the strong DEGs that have been found in this study but not previously implicated in regeneration). This will be important for putting the “conservation”, that is the emphasis of the paper, into context, and will also suggest how regeneration pathways have evolved.

Please refer the response to this comment in the first section

Fig 4B – Not convinced this figure is needed. If kept, figure legend should include the names of clusters 6-10 since they are numbered in the figure.

We have now included the gene ontology on a larger group of DEG and included this in Figure 3.

117 – need close parentheses

This is now fixed

257 – edit – missing/wrong word

NLn 215: This is now fixed

341-343 – edit for grammar

NLn 429: Serine proteases are now discussed in reference to other species

354-356 – unclear – tradeoffs don't arise from impaired differentiation.

NLn 406: we have changed this to “This points to the idea that there may be direct tradeoffs between scarring and regenerative success”

414 – what third upregulation? this was not clearly described prior to this point

NLn 464: The statistically significant regulatory changes between time intervals has now been annotated along the X axis of Figure 4. There was initially a typo and it should have been 24-48hpb instead of 12-48hpb.

463 – prologues – odd word choice

NLn 505: This was a typo and has been changed to “prolongs”

491 – remove “signaling”

NLn 377: This is now fixed

506 – regeneration could be both adaptive and inherited. Clarify.

NLn 482: We have clarified that individual components of the process could be adaptive or inherited

Figure Updates

B.

All of the components of the bioinformatic pipeline have been aligned

Gene Ontology of NOISeq DEG

This is an addition to the original submission. It shows enriched GO terms for each interval grouped into early, middle and late categories. We have also added a summary panel to show genes associated with these GO terms (Supplementary File 6 includes the full list of genes).

A. Key

B. Transcription factors - GO:0003700

C. Peptidases - GO:0004222 & GO:0003824

D. Tubulins - GO:0005200

We reorganized this figure to show the key at the top right. We added annotating to the X axis to reveal statistically significant regulatory changes as designated by NOISeq.

Injury responsive gene expression in whole-body regeneration

This is an additional figure that serves as a summary of what is currently understood about early response gene and wnt ligand expression in whole body regenerators. We included this figure to emphasize the early diverging position of *M. leidy* and the lack of Wnt ligand genes during regeneration in this species.

Supplement

A. Chitin binding(GO.0008061) - Uncut-10m Up

B. Calcium ion binding(GO.0005509) - Uncut-10m Down

C. G-protein coupled receptor activity (GO: 0004930) - 3h-6h Up

D. Catalytic activity(GO:0003824) - 3h-6h Down

E. Structural molecule activity(GO:0005198) - 6h-12h Up

This is an additional figure that serves to show expression of genes included in highly enriched GO categories in Figure 3. The regulatory designation of each is included

A. NOISeq DEG

B. edgeR DEG

C. EBseqHMM DEG

D. Consensus DEG

This figure is a modification of multiple figures in the first submission. Here, we include the top differentially expressed genes from each method and the Venn diagram to show the consensus list

This is an additional figure in which the entirety of the consensus DEG list (118) is displayed on a heatmap. Clusters were generated from this heatmap and the dendrogram is annotated to show where these clusters are designated. Line graphs showing expression of genes in each cluster are also included.

This is an additional figure that shows all of the 18 alpha tubulin genes found in the *M. leidy* gene models

REVIEWERS' COMMENTS:

Reviewer #2 (Remarks to the Author):

The authors have addressed my criticisms and suggestions and present a significantly improved version of the manuscript. The data on ctenophores (here *M. leidy*) are essential from an evolutionary and functional perspective. Figure 3 provides a good overview of differential gene expression and Figure 6 – lack of expression of Wnt ligands in regenerating ctenophores – sheds new light on animal patterning processes. Overall, this is an important study for a better understanding of the regeneration processes in ancestral metazoans. It deserves rapid publication.

Reviewer #3 (Remarks to the Author):

My comments have been addressed well in this revision. A few remaining points:

1) The new wnt section on pg 28 is a great addition. But note that the new text is poorly edited. Please revise:

- Ln 511: Edit. The word “both” is used before a list of three.
- Ln 514: Transition word is missing – add “Although”, “but” or the like.
- Ln 515: The authors have not demonstrated lack of Wnt ligand expression, only that Wnt ligands are not among the significant DEGs. Edit.
- Ln 516: Awkward sentence.

2) The new Fig 6 is a great addition. But note:

- Fig 6: Gene regulatory network part of the visual legend (upper right of figure): The meaning of the “slash” symbol used between “early response gene” and “Wnt” is unclear. It could refer specifically to a direct regulatory relationship between the two, or that one is upstream of the other, or that one or the other is involved. Somehow clarify the meaning in the visual legend and/or in the text of the figure legend itself.
 - Fig 6: Less vertical space is needed between Echinodermata and Platyhelminthes.
 - Fig 6: Ctenophora data summary in lower right: Does “Not DEG” refer to any time point sampled? Or specifically to 3-6hpb? Specify either in the figure or its text legend.
- Fig 6: Acoela is misspelled.

Also:

- Ln 116: extracted rather than extracting.

Reviewer #2 (Remarks to the Author):

The authors have addressed my criticisms and suggestions and present a significantly improved version of the manuscript. The data on ctenophores (here *M. leidy*) are essential from an evolutionary and functional perspective. Figure 3 provides a good overview of differential gene expression and Figure 6 – lack of expression of Wnt ligands in regenerating ctenophores – sheds new light on animal patterning processes. Overall, this is an important study for a better understanding of the regeneration processes in ancestral metazoans. It deserves rapid publication.

Reviewer #3 (Remarks to the Author):

My comments have been addressed well in this revision. A few remaining points:

1) The new wnt section on pg 28 is a great addition. But note that the new text is poorly edited. Please revise:

- Ln 511: Edit. The word “both” is used before a list of three.

This is now fixed

- Ln 514: Transition word is missing – add “Although”, “but” or the like.

“but” has now been added as a transition

- Ln 515: The authors have not demonstrated lack of Wnt ligand expression, only that Wnt ligands are not among the significant DEGs. Edit.

This has been changed to “The lack of differential upregulation of Wnt ligand genes suggests that ctenophore regeneration may differ substantially from other animal groups. Alternatively, while early response genes are widely conserved, they may be regulating the downstream gene expression of other ligands in *M. leidy*.”

- Ln 516: Awkward sentence.

See above edit

2) The new Fig 6 is a great addition. But note:

- Fig 6: Gene regulatory network part of the visual legend (upper right of figure): The meaning of the “slash” symbol used between “early response gene” and “Wnt” is unclear. It could refer specifically to a direct regulatory relationship between the two, or that one is upstream of the other, or that one or the other is involved. Somehow clarify the meaning in the visual legend and/or in the text of the figure legend itself.

GRN notation has been added to the visual legend

- Fig 6: Less vertical space is needed between Echinodermata and Platyhelminthes.

This is now fixed

- Fig 6: Ctenophora data summary in lower right: Does “Not DEG” refer to any time point sampled? Or specifically to 3-6hpb? Specify either in the figure or its text legend.

This has been clarified in the visual legend as well as the text legend

Fig 6: Acoela is misspelled.

This is now corrected

Also:

- Ln 116: extracted rather than extracting.

This is now fixed